# A Private Approximation of the 2nd-Moment Matrix of Any Subsamplable Input

**Bar Mahpud**
Faculty of Engineering
Bar-Ilan University
Israel
mahpudb@biu.ac.il

**Or Sheffet**
Faculty of Engineering
Bar-Ilan University
Israel
or.sheffet@biu.ac.il

## Abstract

We study the problem of differentially private second moment estimation and present a new algorithm that achieve strong privacy-utility trade-offs even for worst-case inputs under subsamplability assumptions on the data. We call an input $(m, \alpha, \beta)$-*subsamplable* if a random subsample of size $m$ (or larger) preserves w.p $\geq 1 - \beta$ the spectral structure of the original second moment matrix up to a multiplicative factor of $1 \pm \alpha$. Building upon subsamplability, we give a recursive algorithmic framework similar to Kamath et al. (2019) that abides zero-Concentrated Differential Privacy (zCDP) while preserving w.h.p the accuracy of the second moment estimation upto an arbitrary factor of $(1 \pm \gamma)$. We then show how to apply our algorithm to approximate the second moment matrix of a distribution $\mathcal{D}$, even when a noticeable *fraction* of the input are outliers.

## 1 Introduction

Estimating the second moment matrix (or equivalently, the covariance matrix) of a dataset is a fundamental task in machine learning, statistics, and data analysis. In a typical setting, given a dataset of $n$ points in $\mathbb{R}^d$, one aims to compute an empirical second moment (or covariance) matrix that is close, in spectral norm, to the true second moment matrix. However, as modern datasets increasingly contain sensitive information, maintaining strong privacy guarantees has become a key consideration.

A natural way to protect sensitive data is through *differential privacy* (DP). In this paper, we focus on the zero-Concentrated Differential Privacy (zCDP) framework (Bun & Steinke, 2016), which offers elegant composition properties and somewhat tighter privacy-utility trade-offs compared to traditional $(\epsilon, \delta)$-DP. While there have been works regarding the estimation of the second moment matrix (and PCA), they mostly focused on Gaussian input or well-conditioned input (see Related Work below). In contrast, our work focuses on a general setting, where the input's range is significantly greater than $\lambda_{\min}$, the least eigenvalue of the 2nd-moment matrix.

Suppose indeed we are in a situation where the first and least eigenvalues of the input's 2nd moment matrix are very different. By and large, this could emanate from one of two options: either it is the result of a few outliers, in which case it is unlikely to approximate the 2nd moment matrix well with DP; or it is the case that the underlying distribution of the input does indeed have very different variances along different axes, and here DP approximation of the input is plausible. Our work is focuses therefore on the latter setting, which we define using the notion of *subsamplability*. Namely that from a sufficiently large random subsample, one can recover a spectral approximation to the original second moment matrix with high probability. This property resonates with classical matrix-concentration results (namely, matrix Bernstein bounds), yet – as our analysis shows – our subsamplability assumption offers a less nuanced path to controlling the tail behavior of the data. In this work we formalize this notion of subsamplability – which immediately gives a non-private

39th Conference on Neural Information Processing Systems (NeurIPS 2025).

approximation the data's second moment matrix, and show how it can be integrated into a privacy-preserving algorithm with some overhead.

**Subsamplability Assumption.** Throughout this paper, we assume that our $n$-size input dataset is subsamplable, which we formally define as follows.

**Definition 1.1.** $((m, \alpha, \beta)$-subsamplability) Let $X \subseteq \mathbb{R}^d$ be a dataset of $n$ points. Fix $m \leq n, \alpha > 0, \beta \in (0, 1)$. Let $\hat{X}_1, \ldots, \hat{X}_{m'}$ be a random subsample of $m' \geq m$ points i.i.d from $X$. Denote $\Sigma = \frac{1}{n} \sum_{i \in [n]} X_i X_i^T$ and $\hat{\Sigma} = \frac{1}{m'} \sum_{i \in [m']} \hat{X}_i \hat{X}_i^T$, then the dataset $X$ is $(m, \alpha, \beta)$-*subsamplable* if:

$$\forall m' \geq m: \qquad \Pr[(1 - \alpha)\Sigma \preceq \hat{\Sigma} \preceq (1 + \alpha)\Sigma] \geq 1 - \beta$$

By assuming subsamplability, we ensure that the critical spectral properties of the data are retained, enabling efficient and accurate private estimation. This assumption provides a tractable way to manage the inherent complexity of the problem while maintaining robustness to variations in the data. On the contrapositive — when the data isn't subsamplable, estimating the second moment matrix becomes significantly more challenging. Furthermore, in the case where the $n$ input points are drawn i.i.d. from some distribution (the case we study in Section 4) subsamplability follows directly from the convergence of a large enough sample to the true (distributional) second moment matrix; alternatively, sans subsamplability we cannot estimate the distribution's second moment matrix.

It is important to note that our subsamplability assumption is *weaker* than standard concentration bounds, which state that *for any* $\alpha, \beta$ there exists $m(\alpha, \beta)$ such that a random subsample of $m$ (or more) points preserve w.p.$\geq 1 - \beta$ the spectral structure of $\Sigma$ upto a $(1 \pm \alpha)$-factor. Here we only require that for *some* $\alpha, \beta$ there exists such a $m(\alpha, \beta)$, a distinction that allows us to cope even with a situation of a well-behaved distribution with outliers, as we discuss in Section 4. In our analysis we require that $\alpha = O(1)$ (we set it as $\alpha \leq 1/2$ purely for the ease of analysis); however, our analysis does require a bound on the $\beta$ parameter, namely having $\beta = O(\alpha/\log(R))$, with $R$ denoting the bound on the $L_2$ norm of all points. It is an interesting open problem to replace this requirement on $\beta$ with a $O(1)$-bound as well. It is also important to note that our result is *stronger* than the baseline we establish: subsamplability implies that using (roughly) $m/\epsilon$ samples, one can succesfully apply the subsample-and-aggregate framework (Nissim et al., 2007) and obtain a $1 \pm O(\alpha)$-approximation of the spectrum of the second moment matrix. In contrast, our algorithm can achieve a $(1 \pm \gamma)$-approximation of the second moment matrix of the "nice" portion of the input even for $\gamma \ll \alpha$ (provided we have enough input points). Details below.

**Contributions.** First, we establish a *baseline* for this problem. Under our subsamplability assumption, we use an off-the-shelf algorithm of Ashtiani & Liaw (2022) that follows the "subsample-and-aggregate paradigm (Nissim et al., 2007) to privately return a matrix $\tilde{\Sigma}$ satisfying $(1 - 2\alpha)\Sigma \preceq \tilde{\Sigma} \preceq (1 + 2\alpha)\Sigma$.

We then turn our attention to our algorithm, which is motivated by the same recursive approach given in Kamath et al. (2019): In each iteration we deal with an input $X$ whose 2nd moment matrix satisfy $I \preceq \Sigma \preceq \kappa I$, add noise (proportional to $\kappa/m$) to its 2nd-moment matrix and find the subspace of large eigenvalue (those that are greater than $\psi\kappa$ for $\psi \approx 1/m$), and then apply a linear transformation $\Pi$ reducing the projection onto the subspace of large eigenvalues by $1/2$, thereby reducing the second moment matrix of $\Pi X$ so that it is $\preceq \frac{3\kappa}{7} I$. So we shrink $R$, the range of the input, to $\sqrt{3/7}R$ and continue by recursion. Yet unlike Kamath et al. (2019) who work with the underlying assumption that the input is Gaussian, we only know that our input is subsamplable, and so in our setting there could be input points whose norm is greater than $\sqrt{3/7}R$ after applying $\Pi$ and whose norm we must shrink to fit in the $\sqrt{3/7}R$-ball. So the bulk of our analysis focuses on these points that undergo shrinking, and show that they all must belong to a particular set we refer to as $P_{\text{tail}}$ (see Definition 3.2). We argue that there aren't too many of them (just roughly a $\beta/m$-fraction of the input) and that even with shrinking these points, the second moment matrix of the input remains $\succeq I$. This allows us to recurse all the way down to a setting where $\kappa \propto m$, where all we have to do is to simply add noise to the 2nd moment matrix to obtain a $(1 \pm \gamma)$-approximation w.h.p.

We then apply our algorithm to an ensemble of points drawn from a general distribution (even a *heavy-tailed* one). So next we consider any distribution $\mathcal{D}$ with a finite second moment $\Sigma_\mathcal{D}$ where

the vector $y = \Sigma_{\mathcal{D}}^{-1/2} x$ for $x \sim \mathcal{D}$ exhibits particular bounds (See Claim 4.2 for further details), and give concrete sample complexity bounds for our algorithm to approximate $\Sigma_{\mathcal{D}}$ up to a factor of $1 \pm \gamma$ w.p. $1 - \xi$. We then consider a mixture of such a well-behaved $\mathcal{D}$ with an $\eta$-fraction of *outliers*. We show that our algorithm allows us to cope with the largest fraction of outliers (roughly $\tilde{O}(1/d)$) provided that the second moment matrix of the outliers $\Sigma_{\text{out}}$ satisfies $\Sigma_{\text{out}} \preceq O(1/\eta)\Sigma_{\mathcal{D}}$. In contrast, the subsample and aggregate baseline (and other baselines too) not only requires a smaller bound on $\eta$ but also has a significantly large sample complexity bound. Details appear in Section 4.

**Organization.** After surveying related work in the remainder of Section 1, we introduce necessary definitions and background in Section 2. In Section 3.1 we survey the baseline of Ashtiani & Liaw (2022), and in Section 3.2 we discuss using existing algorithms to estimate the initial parameters of the input we require, namely its range $R$ and its least eigenvalue $\lambda_{\min}$. Multiplying $R^2$ by $1/\lambda_{\min}$ we obtain an input that indeed satisfies $I \preceq \Sigma \preceq \kappa I$ (with $\kappa = R^2/\lambda_{\min}$). Then, in Section 3.3 we present our algorithm and state its utility theorem, which we prove in Section 3.4. Finally, Section 4 illustrates how to apply our framework to a general (potentially heavy tailed) distribution, including the case of a noticeable fraction of outliers.

**Related Work.** Differential privacy has been extensively studied in the context of mean and covariance estimation, particularly in high-dimensional regimes. Early work by Dwork et al. (2014) proposed private PCA for worst-case bounded inputs via direct perturbation of the second moment matrix, laying foundational tools for differentially private matrix estimation. Subsequently, Nissim et al. (2007) introduced the subsample-and-aggregate framework, which has since become a standard paradigm for constructing private estimators under structural assumptions.

A significant body of research has focused on learning high-dimensional Gaussian distributions under differential privacy. Kamath et al. (2019) introduced a recursive private preconditioning technique for Gaussian and product distributions, achieving nearly optimal bounds while relying on the assumption of a well-behaved (Gaussian) input. Their approach underlies several subsequent advances in private estimation. Building on these ideas, Kamath et al. (2022) proposed a polynomial-time algorithm for privately estimating the mean and covariance of unbounded Gaussians. Their algorithm, which incorporated a novel private preconditioning step, improved both accuracy and computational efficiency.

Ashtiani & Liaw (2022) proposed a general framework that reduces private estimation to its non-private analogue. This yielded efficient, approximate-DP estimators for unrestricted Gaussians with optimal (up to logarithmic factors) sample complexity. Their method also demonstrated the power of reduction-based techniques in bridging private and non-private statistics. Aden-Ali et al. (2021) gave near-optimal bounds for agnostically learning multivariate Gaussians under approximate DP, while Amin et al. (2019) and Dong et al. (2022) revisited the task of private covariance estimation under $\epsilon$-DP and zCDP, respectively. These works introduced trace- and tail-sensitive algorithms for better handling of data heterogeneity.

Recent work has emphasized robustness and practical applicability. For example, Biswas et al. (2020) introduced a robust and accurate mean/covariance estimator for sub-Gaussian data, and Kothari et al. (2022) developed a robust, polynomial-time estimator resilient to adversarial outliers. Further, Alabi et al. (2023) presented near-optimal, computationally efficient algorithms for privately estimating multivariate Gaussian parameters in both pure and approximate DP models.

A particularly notable contribution is by Brown et al. (2023), who studied the problem of differentially private covariance-aware mean estimation under sub-Gaussian assumptions. They introduced a polynomial-time algorithm that achieves strong Mahalanobis distance guarantees with nearly optimal sample complexity. Their techniques also extend to distribution learning tasks with provable guarantees on total variation distance.

Our algorithm outperforms prior methods that rely on per-point bounded leverage and residual conditions—most notably the private covariance estimation algorithm of Brown et al. Brown et al. (2023)—in settings where the dataset may contain a small fraction of outliers or where individual points may exhibit high leverage scores, but the *global spectral structure* is preserved in random subsamples. Unlike their algorithm, which requires strong uniform constraints on every data point (i.e., no large leverage scores), our method only assumes a subsamplability condition that holds *with high probability* over random subsamples. This allows us to tolerate the presence of many multiple

outliers, provided they do not dominate the overall spectrum. Moreover, our algorithm is tailored for *second moment estimation*, and achieves strong utility guarantees even when the second moment matrix has a large condition number – a regime where the estimator of Brown et al. (2023) may incur significant error with the presence of outlier correlated with the directions of small eigenvalues. A more elaborated discussion demonstrating this setting appears in Section 4.2.

## 2   Preliminaries

Throughout the paper, we assume that our instance of dataset is subsamplable, as given in Definition 1.1.

**Notations.**   Let $\mathcal{S}^{d-1}$ denote The *unit sphere* in $\mathbb{R}^d$, which is defined as the set of all points in $d$-dimensional Euclidean space that have unit norm, i.e., $\mathcal{S}^{d-1} = \{x \in \mathbb{R}^d \mid \|x\|_2 = 1\}$. Here, the superscript $d - 1$ indicates that the unit sphere is an object of intrinsic dimension $d - 1$ embedded in $\mathbb{R}^d$.

Let $\mathrm{GUE}(\sigma^2)$ denote the distribution over $d \times d$ symmetric matrices $N$ where for all $i \leq j$, we have $N_{ij} \sim \mathcal{N}(0, \sigma^2)$ i.i.d.. From basic random matrix theory, we have the following guarantee.

**Fact 2.1** (see e.g. Tao (2012) Corollary 2.3.6)**.** *For $d$ sufficiently large, there exist absolute constants $C, c > 0$ such that:* $\Pr\limits_{N \sim \mathrm{GUE}(\sigma^2)}[\|N\|_2 > A\sigma\sqrt{d}] \leq Ce^{-cAd}$ *for all $A \geq C$.*

**Definition 2.2** (Differential Privacy (Dwork et al., 2006))**.** *A randomized algorithm $\mathcal{A}$ satisfies $(\epsilon, \delta)$-differential privacy if, for all datasets $D$ and $D'$ differing in at most one element, and for all measurable subsets $S$ of the output space of $\mathcal{A}$, it holds that:*

$$\Pr[\mathcal{A}(D) \in S] \leq e^\epsilon \Pr[\mathcal{A}(D') \in S] + \delta.$$

**Definition 2.3** (Zero-Concentrated Differential Privacy (zCDP) (Bun & Steinke, 2016))**.** *A randomized algorithm $\mathcal{A}$ satisfies $\rho$-zero-concentrated differential privacy ($\rho$-zCDP) if, for all datasets $D$ and $D'$ differing in at most one element, and for all $\alpha > 1$, the Rényi divergence of order $\alpha$ between the output distributions of $\mathcal{A}$ on $D$ and $D'$ is bounded by $\rho\alpha$, i.e., $D_\alpha(\mathcal{A}(D)\|\mathcal{A}(D')) \leq \rho\alpha$. Here, $\rho \geq 0$ is the privacy parameter that controls the trade-off between privacy and utility, and $D_\alpha$ denotes the Rényi divergence of order $\alpha$.*

**Theorem 2.4** (Bun & Steinke (2016))**.** *If a randomized algorithm $\mathcal{A}$ satisfies $\rho$-zero-concentrated differential privacy ($\rho$-zCDP), then $\mathcal{A}$ also satisfies $(\epsilon, \delta)$-differential privacy for any $\delta > 0$, where:* $\epsilon = \rho + \sqrt{2\rho \ln\left(\frac{1}{\delta}\right)}$.

**Theorem 2.5** (Composition Theorem for $\rho$-zCDP)**.** *Let $\mathcal{M}_1$ and $\mathcal{M}_2$ be two independent mechanisms that satisfy $\rho_1$-zCDP and $\rho_2$-zCDP, respectively. Then their composition $\mathcal{M}_1 \circ \mathcal{M}_2$ satisfies $(\rho_1 + \rho_2)$-zCDP.*

## 3   Technical Analysis

### 3.1   Baseline

In this section, we provide a baseline for the problem of 2nd-moment estimation using subsample and aggregate framework (Nissim et al., 2007). For the lack of space we move the entire discussion of the baseline to Appendix A, and only cite here the conclusion.

**Theorem 3.1.** *Let $\xi$, $\epsilon$, $\delta$ be parameters, and let $X \subseteq \mathbb{R}^d$ be a $(m, \alpha, \beta)$-subsamplable set of $n$ points. Then, there exists an algorithm for which the following properties hold:*

1. *The algorithm is $(2\epsilon, 4e^\epsilon\delta)$-Differential Private.*

2. *The algorithm returns $\tilde{\Sigma}$ satisfying $\|\Sigma^{-1/2}\tilde{\Sigma}\Sigma^{-1/2} - I\| \leq 2\alpha$, where $\Sigma = \frac{1}{n}XX^\top$.*

*These guarantees hold under the following conditions:*

1. *The dataset size satisfies: $n \geq 800m \cdot \max\left\{\sqrt{\frac{2d(d+1/n^2)}{\epsilon}}, \frac{8d\sqrt{\ln(2/\delta)}}{\epsilon}, \frac{8\ln(2/\delta)}{\epsilon}, \frac{12\sqrt{d\ln(2/\delta)}}{\epsilon\eta}, \frac{\ln\left(1+\frac{e^\epsilon-1}{2\delta}\right)}{80\epsilon}\right\}$ where $\eta = \frac{\alpha}{48C(\sqrt{d}+\sqrt{\ln(4/\xi)})}$ for a sufficiently large constant $C > 0$.*

*2. The subsamplability parameters satisfy $m \geq 2\beta n/\xi$.*

In particular, Item 1 suggests a sample complexity bound of $n = \Omega(\frac{d \cdot m(\alpha, \beta)}{\epsilon \alpha})$.

## 3.2 Finding Initial Parameters

Our recursive algorithm requires as input two parameters that characterize the "aspect ratio" of the input, namely $-R_{\max}$, the maximum distance of any point from the origin, and $\lambda_{\min}$, the minimum eigenvalue of the input. These two parameters give us the initial bounds, as they imply that $\lambda_{\min} I \preceq \Sigma \preceq R_{\max}^2 I$. Due to space constraints, we move the entire discussion, regarding how to apply off-the-shelf algorithms, or modify such algorithms, to obtain these initial parameters to Appendix B.

## 3.3 Main Algorithm and Theorem

Next, we detail our algorithm that approximates the second moment of the input. Its starting point is the assumption that the input has a known bound on the $L_2$-norm of each point $R$, and that the second moment matrix of the input, $\Sigma$, satisfies $I \preceq \Sigma \preceq R^2 I$.

---

**Algorithm 1** DP Second Moment Estimation

**Input:** a $(m, \alpha, \beta)$-subsamplable set of $n$ points $X \subseteq \mathbb{R}^d$, parameters: error parameter $\xi \in (0, 1)$, privacy parameter $\rho$, covering radius $R$.

1: Set: $\eta \leftarrow 1/2$, $\quad T \leftarrow \log_{7/3}\left(\frac{(\frac{1}{1-\alpha})R^2}{640m}\right)$, $\quad \psi \leftarrow \frac{1}{10m}, c \leftarrow \frac{1}{80m}$, $\quad C \leftarrow 640m$, $\quad \kappa \leftarrow R^2$

2: $\tilde{\Sigma} \leftarrow$ RecDPSME $\left(\sqrt{(\frac{1}{1-\alpha})}X, \eta, \psi, C, c, (\frac{1}{1-\alpha})\kappa, \sqrt{(\frac{1}{1-\alpha})}R, T, \xi, \rho\right)$

3: **return** $(1 - \alpha)\tilde{\Sigma}$

---

**Algorithm 2** Recursive DP Second Moment Estimation (RecDPSME)

**Input:** a set of $n$ points $X \subseteq \mathbb{R}^d$, parameters: linear shrinking $\eta < 1$, eigenvalue threshold $\psi < 1$, stopping value C, noise $c$, eigenvalue upper bound $\kappa$, radius $R$, iterations bound $T$, error parameter $\xi$, privacy loss $\rho$.

1: Set $\sigma \leftarrow \frac{4R^2\sqrt{T}}{n\sqrt{2\rho}}$

2: Sample $N \sim \text{GUE}(\sigma^2)$

3: $\tilde{\Sigma} \leftarrow \frac{1}{n}XX^T + N$

4: **if** $\kappa \leq C$ **then**

5: $\quad$ **return** $\tilde{\Sigma}$.

6: **end if**

7: $V \leftarrow \text{Span}(\{v_i : \text{eigenvector of } \tilde{\Sigma} \text{ with eigenvalue } \geq \psi\kappa\})$

8: $\Pi \leftarrow \eta\Pi_V + \Pi_{V^\perp}$ $\qquad\qquad\qquad$ {$\Pi_U$ denotes the projection matrix onto $U$.}

9: $Y \leftarrow \sqrt{\frac{8}{7}}\Pi X$.

10: $X_{next} \leftarrow S(Y)$ $\qquad\qquad$ where $S(Y) = \left\{y \cdot \min\left\{1, \sqrt{\frac{\frac{3}{7}R}{\|y\|^2}}\right\} : y \in Y\right\}$

11: $\Sigma_{\text{rec}} \leftarrow$ RecDPSME $\left(X_{next}, \eta, \psi, C, c, \frac{3}{7}\kappa, \sqrt{\frac{3}{7}}R, T, \xi, \rho\right)$

12: **return** $\frac{7}{8}\Pi^{-1}\Sigma_{\text{rec}}\Pi^{-1}$

---

In our analysis, the following definition plays a key role.

**Definition 3.2.** Let $X$ be a $(m, \alpha, \beta)$-subsamplable set. We denote $P_{\text{tail}}$ as the set of points whose projection onto some direction $u$ in $\mathbb{R}^d$ is $m$ times greater than expected, namely

$$P_{\text{tail}} = \left\{x \in X : \exists u \in \mathbb{R}^d : \langle x, u\rangle^2 > m(1 + \alpha) \cdot \frac{1}{n}\sum_{x \in X}\langle x, u\rangle^2\right\}$$

**Theorem 3.3.** *Fix parameters $\xi \in (0,1)$, $\rho > 0, \gamma > 0$ and $\kappa \geq 1$. Let $X \subset \mathbb{R}^d$ be a $(m, \alpha, \beta)$-subsamplable set of $n$ points bounded in $L_2$ norm by $R^2$, with $\alpha \leq 1/2$ and $\beta \leq \frac{\alpha}{4(1+\alpha)\log(\frac{R^2}{(1+\alpha)m})}$*

*s.t. $I \preceq \Sigma \preceq R^2 I$ where $\Sigma = \frac{1}{n}XX^T$. Then, denoting $T = \log_{7/3}\left(\frac{(\frac{1}{1-\alpha})R^2}{640m}\right) = O(\log(R/m))$, we*

*have that Algorithm 1 satisfies $\rho$-zCDP,[1] and if*

$$n \geq \Omega\left(m\sqrt{\frac{d}{\rho}}\left(\sqrt{T}\log(T/\xi) + \frac{\log(1/\xi)}{\gamma}\right)\right)$$

*Then (1) $P_{tail}$ holds at most a $(\frac{\beta+\beta^2}{m})$-fraction of $|X|$, and (2) w.p. $\geq 1 - 2\xi$ it outputs $\tilde{\Sigma}$ such that:*

$$(1-\gamma)\Sigma_{eff} \preceq \tilde{\Sigma} \preceq (1+\gamma)\Sigma$$

*where $\Sigma_{eff} = \frac{1}{n}\sum_{x \in X \setminus P_{tail}} xx^T$.*

### 3.4 Algorithm's Analysis

Next we prove Theorem 3.3. Momentarily we shall argue that Algorithm 2 repeats for at most $T = \log_{7/3}\left(\frac{(\frac{1}{1-\alpha})R^2}{640m}\right)$ iterations. Based on Fact 2.1 and on the bound on the number of iterations, it is simple to argue that the following event holds w.p. $\geq 1 - \xi$:

$$\mathcal{E} := \text{in each iteration of Algorithm 2 we have } \|N\| \leq cR^2 = c\kappa$$

which follows from the fact that in each iteration the upper bound on the largest eigenvalue of $\Sigma$ is at most $\kappa = R^2$. We continue our analysis conditioning on $\mathcal{E}$ holding.

The analysis begins with the following lemma, that shows that under $\mathcal{E}$ we have that in each iteration the eigenvalues of $\Sigma$ decrease. Its proof is very similar to the proof in Kamath et al. (2019) and so it is deferred to Appendix C.

**Lemma 3.4.** *Given $X = \{X_1, ..., X_n\} \subset \mathbb{R}^d$, $C > 0, c > 0, 0 < \eta < 1, 0 < \psi < 1$, and $\kappa \geq 1$ s.t. $I \preceq \Sigma \preceq \kappa I$ where $\Sigma = \frac{1}{n}XX^T$. Let $V \leftarrow \text{Span}(\{v_i : \lambda_i \geq \psi\kappa\})$ of the largest eigenvalues of the noisy $\tilde{\Sigma}$ and let $\Pi = \eta\Pi_V + \Pi_{V^\perp}$. Given that: $n \geq \Omega\left(m\sqrt{\frac{dT}{\rho}}\ln(T/\xi)\right)$ Then:*

$$\left(1 - \frac{1}{\eta^2\psi - c} \cdot \frac{1}{\kappa}\right)I \preceq \Pi\Sigma\Pi \preceq (\eta^2 + \psi + 2c)\kappa I$$

*In particular, if $\kappa > C$ for some $C$ then: $I \preceq \frac{1}{(1-\frac{1}{\eta^2\psi - c})}\Pi\Sigma\Pi \preceq \frac{\eta^2+\psi+2c}{(1-\frac{1}{C(\eta^2\psi - c)})}\kappa I$.*

**Corollary 3.5.** *Given $X = \{X_1, ..., X_n\} \subset \mathbb{R}^d$ and $\kappa \geq 1$ s.t. $I \preceq \Sigma \preceq \kappa I$ where $\Sigma = \frac{1}{n}XX^T$. Let $V, \Pi$ be as in Algorithm 2, and set $\eta = 1/2$, $\psi = 1/10m$, $c = 1/80m$ and $C = 640m$ in Lemma 3.4. Then w.p. $\geq 1 - \xi$:*

$$\left(1 - 80m \cdot \frac{1}{\kappa}\right)I \preceq \Pi\Sigma\Pi \preceq \left(\frac{1}{4} + \frac{1}{8m}\right)\kappa I$$

*In particular, if $\kappa > C$ then: $I \preceq \frac{8}{7}\Pi\Sigma\Pi \preceq \frac{3}{7}\kappa I$.*

Based on Corollary 3.5 we can bound the number of iterations of the algorithm.

**Corollary 3.6.** *Algorithm 2 has $T = \log_{7/3}\left(\frac{(\frac{1}{1-\alpha})R^2}{640m}\right)$ iterations.*

*Proof.* The algorithm halts when $\left(\frac{3}{7}\right)^T\left(\frac{1}{1-\alpha}\right)\kappa \leq C = 640m$ so: $T \geq \log_{3/7}\left(\frac{640m}{(\frac{1}{1-\alpha})\kappa}\right) = \log_{7/3}\left(\frac{(\frac{1}{1-\alpha})R^2}{640m}\right)$. $\qquad\square$

---

[1] Note that the privacy of Algorithm 1 holds for any input $X$ with bounded $L_2$-norm, regardless of $X$ being subsamplable or not.

With $\Pi$ reducing the largest eigenvalues of $\Sigma$ from $\kappa$ to $3\kappa/7$, we now proceed and bound the radius of all datapoints by from $R$ to $\sqrt{3/7}R$. This is where our analysis diverges from the analysis in Kamath et al. (2019). Whereas Kamath et al rely on the underlying Gaussian distribution to argue they have no outliers, we have to deal with outliers. For our purpose, a datapoint $x$ is an outlier if the shrinking function $S$ (Step 10 in Algorithm 2) reduces the norm of $\Pi x$ since $\|\Pi x\| > \sqrt{3/7}R$. In the following claims we argue that all outliers lie in $P_{\text{tail}}$ (Definition 3.2), and moreover, that by shrinking the outliers we do not alter the second moment matrix all too much. We begin by arguing that there aren't too many outliers.

**Claim 3.7.** Analogously to Definition 3.2, fix any $m' \geq m$ and define $P_{\text{tail}}(m') = \{x \in X : \exists u \in \mathbb{R}^d : (x^T u)^2 > m'(1+\alpha)\frac{1}{n}\sum_i (x_i^T u)^2\}$. Then it holds that

$$\Pr_{x \in_R X}[x \in P_{\text{tail}}(m')] \leq \frac{\beta + \beta^2}{m'}.$$

*Proof sketch.* The proof applies the $(m, \alpha, \beta)$-subsamplability property: if a point violates the bound, it would contradict subsamplability with non-negligible probability. A simple union bound and tail approximation then yield the claimed bound. Full details are deferred to Appendix C.2. □

**Lemma 3.8.** Let $X$ be a $(m, \alpha, \beta)$-subsamplable with $\beta \leq \frac{\alpha}{4(1+\alpha)\log(\frac{R^2}{(1+\alpha)m})}$. Let $P = X \setminus P_{tail}$. Then:

$$\forall u \in \mathbb{R}^d : \frac{1}{n}\sum_{x \in P}\langle x, u\rangle^2 \geq (1-\alpha)\frac{1}{n}\sum_{x \in X}\langle x, u\rangle^2$$

*Proof sketch.* We partition the tail points according to the magnitude of their contribution and apply Claim 3.7 to bound the measure of each bucket. Summing across buckets shows that the overall loss from removing the tail points is small. Full proof is deferred to Appendix C.3. □

**Lemma 3.9.** At each iteration $t$ of Algorithm 2, only points belonging to $P_{\text{tail}}$ are subjected to shrinking, given that $\alpha < 1/2$, $\psi = \frac{1}{10m}$ and $c = \frac{1}{80m}$.

*Proof sketch.* The proof uses induction over iterations. Shrinking happens only if a point's mass in a low-eigenvalue subspace is too large. By carefully tracking how shrinking operates and applying Weyls theorem and Lemma 3.8, we show that only initially bad points (i.e., those in $P_{\text{tail}}$) can cause such violations. Full proof appears in Appendix C.4. □

**Corollary 3.10.** In all iterations of the algorithm it holds that $\Sigma \succeq I$, namely, that the least eigenvalue of the second moment matrix of the input is $\geq 1$.

*Proof sketch.* We argue by induction that removing or shrinking tail points preserves a spectral lower bound. Using Lemma 3.8 and the shrinkage structure from Lemma 3.9, the transformation at each step maintains the least eigenvalue above 1. Full proof is provided in Appendix C.5. □

*Proof of Theorem 3.3.* First we argue that Algorithm 1 is $\rho$-zCDP. Given two neighboring data sets $X, X'$ of size $n$ which differ in that one contains $X_i$ and the other contains $X_i'$, the covariance matrix of these two data sets can change in Spectral norm by at most:

$$\|\frac{1}{n}(X_i X_i^T - X_i' X_i'^T)\|_2 \leq \frac{1}{n}\|(X_i - X_i')(X_i - X_i')^T\|_2 \leq \frac{1}{n}\|(X_i - X_i')\|_2\|(X_i - X_i')^T\|_2 \leq \frac{(2R)^2}{n}$$

Since Algorithm 1 invokes $T$ calls to Algorithm 2 each preserving $\rho/T$-zCDP, thus the privacy guarantee of Algorithm 1 follows from sequential composition of zCDP.

We now turn to proving the algorithm's utility. From Claim 3.7 we conclude that $|P_{\text{tail}}|$ is indeed at most $(\frac{\beta+\beta^2}{m})$-fraction of $|X|$. We prove by recursion that: $(1-\gamma)\Sigma_{\text{eff}} \preceq \tilde{\Sigma} \preceq (1+\gamma)\Sigma$.

**Stopping Rule:** Let $X^T$ be the input at the final iteration $T$ and let $P = X \setminus P_{\text{tail}}$. Denote $\Sigma(\cdot)$ as the second moment matrix operator. We know that throughout the algorithm, the points from $P$ were not shrunk. Moreover, Corollary 3.10 assures that the least eigen value of $\Sigma(X)$ is $\geq 1$. Additionally, our bound on $n$ yields that when $\kappa \leq C$ then the noise matrix $N$ we add satisfy that $\|N\|_2 \leq \gamma$ w.p. $\geq 1 - \xi$. It thus follows that $(1-\gamma)\Sigma(X^T) \preceq \Sigma + N \preceq (1+\gamma)\Sigma(X^T)$ as required.

**Recursive Step:** Let $X^t$ be the input at iteration $t \leq T$. Then, by Lemma 3.4, we have: $I \preceq \Sigma \left(\frac{8}{7}\Pi X^t\right) \preceq \frac{3}{7}\kappa I$. Lemma 3.9 ensures that $S(\Pi X^t)$ shrinks only points from $P_{\text{tail}}$ and so Corollary 3.10 assures that the eigenvalue is $\geq 1$ throughout the recursive iterations. Hence, by the inductive hypothesis, our recursive call returns $\Sigma_{\text{rec}}$ such that:

$$(1-\gamma)\Sigma_{\text{eff}}\left(\sqrt{\frac{8}{7}}\Pi X^t\right) \preceq \Sigma_{\text{rec}} \preceq (1+\gamma)\Sigma\left(\sqrt{\frac{8}{7}}\Pi X^t\right),$$

which implies:

$$(1-\gamma)\Sigma_{\text{eff}}(X^t) \preceq \frac{7}{8}\Pi^{-1}\Sigma_{\text{rec}}\Pi^{-1} \preceq (1+\gamma)\Sigma(X^t).$$

Proving the required for any intermediate iteration of Algorithm 2. $\qquad\square$

# 4 Applications: Coping with Outliers

## 4.1 Input Drawn from 'Nice' Distributions

First, we show our algorithm returns an approximation of the 2nd-moment matrix when the input is drawn from a distribution $\mathcal{D}$. Throughout this section, we apply the Matrix-Bernestein Inequality.

**Fact 4.1.** Let $Z$ be the sum of $m$ i.i.d. matrices $Z = \sum_i Z_i$, whose mean is 0 and have norm bounded by $\|Z_i\| \leq R$ almost surely. Then, denoting $\sigma^2 = \|\mathbb{E}[ZZ^T]\|$, it holds that

$$\Pr[\|Z\| > t] \leq 2d\exp\left(\frac{-t^2/2}{\sigma^2 + Rt/3}\right)$$

We can apply Fact 4.1 above to measure how well the sample covariance estimator approximates the true covariance matrix of a general distribution using the following claim (proof deferred to Appendix C.6.)

**Claim 4.2.** Let $\mathcal{D}$ be a distribution on $\mathbb{R}^d$ with a finite second moment $\Sigma$. Consider a random vector $y$ chosen by drawing $x \sim D$ and then multiplying $y = \Sigma^{-1/2}x$. Suppose that $\|y\| \leq M_1$ a.s. that we also have a bound $\|\mathbb{E}[(y^Ty)yy^T]\| \leq M_2$. Fix $\alpha, \beta > 0$. If we draw $m = \max\left\{\frac{2M_2}{\alpha^2}, \frac{2(1+M_1^2)}{3\alpha}\right\} \cdot \ln(4d/\beta)$ examples from $\mathcal{D}$ and compute the empirical second moment matrix $\hat{\Sigma}$, then w.p. $\geq 1 - \beta$ it holds that

$$\|\Sigma^{-1/2}\hat{\Sigma}\Sigma^{-1/2} - I\|_2 \leq \alpha$$

Recall that we $(1 \pm \gamma)$-approximate the 2nd-moment matrix *of the input* w.p. $\geq 1-\xi$. Thus, we need the input itself to be a $(1 \pm \gamma)$-approximation of the 2nd-moment matrix of the distribution. (We can then apply Fact A.3 to argue we get a $1 \pm O(\gamma)$ approximation of the distribution's second moment matrix.) This means our algorithm requires

$$m(\gamma, \xi) + O\left(\frac{m(\alpha, \beta)}{\gamma}\sqrt{\frac{d}{\rho} \cdot \log(\frac{R}{\lambda_{\min}})}\log(\log(\frac{R}{\lambda_{\min}})/\xi)\right) \tag{1}$$

for $\alpha = 1/2$ and $\beta = O(\frac{1}{\log(R/\lambda_{\min})})$ in order to return a $(1 \pm O(\gamma))$-approximation of the 2nd moment of the distribution w.p. $\geq 1 - O(\xi)$. In Appendix D we give concrete examples of distributions for which this bound is applicable, including (bounded) heavy-tail distributions.

## 4.2 Distributional Input with Outliers

Next, we consider an application to our setting, in which we take some well-behaved distribution $\mathcal{D}$ and add to it outliers. Consider $\mathcal{D}$ to be a distribution that for any $\gamma, \xi > 0$ is $m(\gamma, \xi)$-subsamplable for $m = O(\frac{d\ln(d/\xi)}{\gamma^2})$. We consider here inputs that are composed of $(1-\eta)$-fraction of good points and $\eta$-fraction of outliers. We thus denote the second moment matrix of the input as

$$\Sigma = (1-\eta)\Sigma_{\mathcal{D}} + \eta\Sigma_{\text{out}}$$

We assume throughout that the least eigenvalue of $\Sigma_{\mathcal{D}}$ is $\lambda_{\min}$. Our goal is to return, w.h.p. $(\geq 1-\xi)$ an approximation of $\Sigma_{\mathcal{D}}$ using a DP algorithm.

**Inapplicability of Brown et al. (2023).** The work of Brown et al. (2023) shows that if the input has $\lambda$-bounded leverage scores, namely, if $\forall x, x^T(\frac{1}{n}XX^T)^{-1}x \leq \lambda$, then they recover the second moment of the input with $O(\lambda\frac{\sqrt{d}}{\epsilon})$ overhead to the sampling complexity. However, in this case one can set outliers so that their leverage scores is $R^2/\lambda_{\min}$ (provided the input has $L_2$-norm bound of $R$). We argue that the algorithm of Brown et al. (2023) is unsuited for such a case. Indeed, the algorithm of Brown et al. (2023) has an intrinsic "counter" of outliers (referred to as score), which when reached $O(1/\epsilon)$ causes the algorithm to return 'Failure'.[2] So either it holds that $\eta$ is so small that the overall number of outliers is a constant (namely, $\eta n = O(1/\epsilon)$), or we set the bound on the leverage scores to be $R^2/\lambda_{\min}$ and suffer the cost in sample complexity.

**A Private Learner.** Suppose $\eta$ is very small. In this case we can simply take some off-the-shelf $(\epsilon, \delta)$-DP algorithm with sample complexity $m(\gamma, \xi, \epsilon, \delta)$ that approximates the second moment matrix, and run in over a subsample of $m$ points out of that input. In order for this to work we require that $\eta$ would be smaller than $O(\frac{\xi}{m(\gamma, \xi, \epsilon, \delta)})$, so that a subset of size $m$ would be clean of any outliers.

**Subsample and Aggregate.** The framework of Subsample and Aggregate (Nissim et al., 2007) is in a way a 'perfect fit' for the problem: we subsample $t$ datasets of size $m(\gamma, \xi)$ each, and then wisely aggregate the (majority of the) $t$ results into one. However, in order for this to succeed, it is required that most of the $t$ subsamples are clean of outliers. In other words, we require that the probability of a dataset to be clean ought to be $> 1/2$, namely - $(1-\eta)^{m(\gamma, \xi)} > 1/2$ or alternatively that $\eta = O(\frac{1}{m(\gamma, \xi)})$, which in our case means $\eta = O(\frac{\gamma^2}{d\log(d/\xi)})$. We analyze this paradigm as part of the subsample-and-aggregate baseline we establish (Appendix A), and the subsample-and-aggregate baseline requires

$$\tilde{O}\left(\frac{d \cdot m(\gamma, \xi)}{\epsilon\gamma}\right) = \tilde{O}\left(\frac{d^2 \log(d/\beta)}{\epsilon\gamma^3}\right)$$

in order to return a $(1 \pm O(\gamma))$-approximation of the 2nd moment of the distribution w.p. $\geq 1 - O(\xi)$.

**Our Work.** Our work poses an alternative to the above mentioned techniques. Rather than having $n < \frac{1}{m(\gamma, \xi)}$, we have a slightly more delicate requirement. We require that there exists $\alpha = 1/2$ and $\beta \leq \frac{1}{12\log(\frac{R}{\lambda_{\min}})}$ such that $\eta = O(\frac{\beta}{m(\alpha, \frac{\beta}{2})})$. (In particular, for the given $\mathcal{D}$ it implies that we require that $\eta = O(\frac{1}{d\log(d)\log(R/\lambda_{\min})})$, which is considerably higher value than in the case of subsample and aggregate discussed above.) This way, we can argue that w.p. $\geq 1 - \frac{\beta}{2}$ it holds that a subsample of size $m(\alpha, \frac{\beta}{2})$ contains only points from $\mathcal{D}$ and that w.p. $\geq 1 - \frac{\beta}{2}$ that sample is 'good' in the sense that its empirical second moment satisfy $\hat{\Sigma} \approx \Sigma_{\mathcal{D}}$.

However, we also require that the subsample of size $m(\alpha, \frac{\beta}{2})$ would satisfy that its empirical second moment matrix $\hat{\Sigma}$ satisfies that $(1 - \frac{1}{2})\Sigma \preceq \hat{\Sigma} \preceq (1 + \frac{1}{2})\Sigma$ since we set $\alpha = \frac{1}{2}$. As $\hat{\Sigma} \approx \Sigma_{\mathcal{D}}$ it follows that it suffices to require that

$$(1 - \frac{1}{8})[(1-\eta)\Sigma_{\mathcal{D}} + \eta\Sigma_{\text{out}}] \preceq \Sigma_{\mathcal{D}} \preceq (1 + \frac{1}{8})[(1-\eta)\Sigma_{\mathcal{D}} + \eta\Sigma_{\text{out}}]$$

Some arithmetic shows that the upper bound is easily satisfied when $\frac{\eta}{1-\eta} \leq \frac{1}{8}$ (which clearly holds for our value of $\eta$), yet the lower bound requires that we have

$$\Sigma_{\text{out}} \preceq (\frac{1}{7\eta} + 1)\Sigma_{\mathcal{D}} = O(1/\eta)\Sigma_{\mathcal{D}}$$

Under these two conditions, our work returns w.p. $1 - O(\xi)$ a matrix $\tilde{\Sigma}$ that satisfies that $\tilde{\Sigma} \succeq (1 - O(\gamma))\Sigma_{\mathcal{D}}$, with sample complexity of

$$O\left(m(\gamma, \xi) + \frac{m(\alpha, \frac{\beta}{2})}{\gamma}\sqrt{\frac{d \cdot \log(R/\lambda_{\min})}{\rho}}\log(\frac{\log(R/\lambda_{\min})}{\xi})\right) = O\left(\frac{d\log(d/\xi)}{\gamma^2} + \frac{d^{3/2}\log(d)\log^{3/2}(R/\lambda_{\min})\log(\frac{\log(R/\lambda_{\min})}{\xi})}{\gamma\sqrt{\rho}}\right)$$

---

[2]Moreover, in their algorithm, this 'score' intrinsically cannot be greater than $k = O(1/\epsilon)$ as they use a particular bound of the form $e^{k/\epsilon}$.

## Acknowledgments and Disclosure of Funding

O.S. is supported by the BIU Center for Research in Applied Cryptography and Cyber Security in conjunction with the Israel National Cyber Bureau in the Prime Ministers Office, and by ISF grant no. 2559/20. Both authors thank the anonymous reviewers for their suggestions and advice on improving this paper.

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

# A   Baseline

In this section, we provide a baseline for the problem of 2nd-moment estimation using subsample and aggregate framework (Nissim et al. (2007)).

In this baseline, we work with the following notion of a convex semimetric space. The key property to keep in mind is that for semimetric spaces, we only have an approximate triangle inequality, as long as the points are *significantly close* to one another.

**Definition A.1.** Let $\mathcal{Y}$ be a convex set and let dist $: \mathcal{Y} \times \mathcal{Y} \to \mathbb{R}_{\geq 0}$. We say $(\mathcal{Y}, \text{dist})$ is a convex semimetric space if there exist absolute constants $t \geq 1$, $\phi \geq 0$, and $r > 0$ such that for every $k \in \mathbb{N}$ and every $Y, Y_1, Y_2, \ldots, Y_k \in \mathcal{Y}$, the following conditions hold:

1. $\text{dist}(Y, Y) = 0$ and $\text{dist}(Y_1, Y_2) \geq 0$.

2. **Symmetry.** $\text{dist}(Y_1, Y_2) = \text{dist}(Y_2, Y_1)$.

3. $t$**-approximate** $r$**-restricted triangle inequality.** If both $\text{dist}(Y_1, Y_2), \text{dist}(Y_2, Y_3) \leq r$, then
$$\text{dist}(Y_1, Y_3) \leq t \cdot (\text{dist}(Y_1, Y_2) + \text{dist}(Y_2, Y_3)).$$

4. **Convexity.** For all $\alpha \in \Delta_k$,
$$\text{dist}\left(\sum_i \alpha_i Y_i, Y\right) \leq \sum_i \alpha_i \text{dist}(Y_i, Y).$$

5. $\phi$**-Locality.** For all $\alpha, \alpha' \in \Delta_k$,
$$\text{dist}\left(\sum_i \alpha_i Y_i, \sum_i \alpha'_i Y_i\right) \leq \sum_i |\alpha_i - \alpha'_i| \left(\phi + \max_{i,j} \text{dist}(Y_i, Y_j)\right).$$

where $\Delta_k$ denotes the $k$-dimensional probability simplex. When $r$ is unspecified, we take it to mean $r = \infty$ and refer to it as a $t$-approximate triangle inequality.

The following technical lemma (whose proof appears in Appendix C in Ashtiani & Liaw (2022)) is helpful for learning second moment matrices.

**Lemma A.2.** Let $\mathcal{S}_d$ be the set of all $d \times d$ positive definite matrices. For $A, B \in \mathcal{S}_d$, let
$$\text{dist}(A, B) = \max\{\|A^{-1/2}BA^{-1/2} - I\|, \|B^{-1/2}AB^{-1/2} - I\|\}.$$

Then $(\mathcal{S}_d, \text{dist})$ is a convex semimetric which satisfies a $(3/2)$-approximate 1-restricted triangle inequality and 1-locality.

Based on Lemma A.2, the following distance function forms a semimetric space for positive definite matrices:

$$\text{dist}(\Sigma_1, \Sigma_2) = \begin{cases} \max(\|\Sigma_2^{-1/2}\Sigma_1\Sigma_2^{-1/2} - I_d\|, \|\Sigma_1^{-1/2}\Sigma_2\Sigma_1^{-1/2} - I_d\|) & \text{if rank}\Sigma_1 = \text{rank}\Sigma_2 = d \\ \infty & \text{otherwise} \end{cases}$$

**Fact A.3.** Let $A, B$ be $d \times d$ matrices and suppose that $\|A^{-1/2}BA^{-1/2} - I\| \leq \gamma \leq 1/2$. Then $\|B^{-1/2}AB^{-1/2} - I\| \leq 4\gamma$

---
**Algorithm 3** Baseline DP Second Moment Estimation
---
**Input:** a set of $n$ points $X \subseteq \mathbb{R}^d$, subsamplability parameters $m, \alpha, \beta$, error parameter $\xi \in (0, 1)$, privacy parameters $\epsilon, \delta$.

1: Randomly split $X$ into $T = \lfloor n/m \rfloor$ subgroups $X_1, \ldots, X_T$ of size $m$.
2: **for** $t \in [T]$ **do**
3:    $\Sigma_t \leftarrow \frac{1}{m} X_t X_t^T$
4: **end for**
5: **for** $t \in [T]$ **do**
6:    $q_t \leftarrow \frac{1}{T} |\{t' \in [T] : \text{dist}(\Sigma_t, \Sigma_{t'}) \leq \frac{2\alpha}{1-\alpha}\}|$
7: **end for**
8: $Q \leftarrow \frac{1}{T} \sum_{t \in [T]} q_t$
9: $Z \sim \text{TLap}(2/T, \epsilon, \delta)$
10: $\tilde{Q} \leftarrow Q + Z$
11: **if** $\tilde{Q} < 0.8 + \frac{2}{T\epsilon} \ln(1 + \frac{e^\epsilon - 1}{2\delta})$ **then**
12:    fail and return $\perp$
13: **end if**
14: **for** $t \in [T]$ **do**
15:    $w_t = \min(1, 10 \max(0, q_t - 0.6))$
16: **end for**
17: $\hat{\Sigma} \leftarrow \sum_{t \in [T]} w_t \Sigma_t / \sum_{t \in [T]} w_t$
18: $N \sim \mathcal{N}(0, 1)_{d \times d}$
19: $\eta \leftarrow \frac{\alpha}{48C(\sqrt{d} + \sqrt{\ln(4/\beta)})}$   {$C$ some large constant}
20: **return** $\tilde{\Sigma} = \hat{\Sigma}^{1/2}(I + \eta N)(I + \eta N)^T \hat{\Sigma}^{1/2}$
---

**Lemma A.4.** (Utility Analysis) Let $\Sigma = \frac{1}{n} X X^T$ and set $\eta = \frac{\alpha}{48C(\sqrt{d} + \sqrt{\ln(4/\xi)})}$ for a sufficiently large constant $C > 0$. Then w.p. $\geq 1 - \xi$ Algorithm 3 returns $\tilde{\Sigma}$ such that $\text{dist}(\Sigma, \tilde{\Sigma}) \leq 2\alpha$ given that:

$$n \geq \frac{10m}{\epsilon} \ln(1 + \frac{e^\epsilon - 1}{2\delta}) \quad \text{and} \quad m \geq \frac{2\beta n}{\xi}$$

*Proof.* Indeed, we have that

$$\|\hat{\Sigma}^{-1/2} \tilde{\Sigma} \hat{\Sigma}^{-1/2} - I\| = \|(I + \eta N)(I + \eta N)^T - I\|$$
$$\leq 2\eta \|N\| + \eta^2 \|NN^T\|$$
$$\leq 2\eta C(\sqrt{d} + \sqrt{\ln(4/\xi)}) + \eta^2 (C(\sqrt{d} + \sqrt{\ln(4/\xi)}))^2$$

where we used the fact that $\|N\| \leq C(\sqrt{d} + \sqrt{\ln(4/\xi)})$ w.p. $\geq 1 - \xi/2$. Applying $\eta$ as defined in the lemma gives that:

$$\|\hat{\Sigma}^{-1/2} \tilde{\Sigma} \hat{\Sigma}^{-1/2} - I\| \leq \alpha/12$$

Following from Fact A.3 we have that:

$$\|\tilde{\Sigma}^{-1/2} \hat{\Sigma} \tilde{\Sigma}^{-1/2} - I\| \leq \alpha/3$$

So we have that $\text{dist}(\tilde{\Sigma}, \hat{\Sigma}) \leq \alpha/3$.

Now we show that $\text{dist}(\Sigma, \hat{\Sigma}) \leq \alpha$ w.p. $\geq 1 - \xi/2$:
Based on the subsamplability assumption, we know that w.p. $\geq (1 - \beta)^T$:

$$\forall t \in [T] : \text{dist}(\Sigma_t, \Sigma) \leq \alpha$$

It means that w.p. $\geq (1 - \beta)^T$ all $t \neq t' \in [T]$ satisfy:

$$\begin{cases} \text{dist}(\Sigma_t, \Sigma) \leq \alpha \\ \text{dist}(\Sigma_{t'}, \Sigma) \leq \alpha \end{cases} \implies \text{dist}(\Sigma_t, \Sigma_{t'}) \leq \frac{2\alpha}{1 - \alpha}$$

Hence w.p. $\geq (1 - \beta)^T \geq e^{-\frac{2\beta n}{m}} \geq 1 - \xi/2$ all $t \in [T]$ satisfy:

$$q_t = \frac{1}{T} \left| \left\{ t' \in [T] : \text{dist}(\Sigma_t, \Sigma_{t'}) \leq \frac{2\alpha}{1 - \alpha} \right\} \right| = 1$$

Finally, $Q = 1 > 0.8 + \frac{2}{T\epsilon} \ln(1 + \frac{e^\epsilon - 1}{2\delta})$ w.p. $\geq 1 - \xi/2$ and therefore the algorithm does not fail w.p. $\geq 1 - \xi$.

Now obviously $\mathrm{dist}(\Sigma, \hat{\Sigma}) \leq \alpha$ since $\hat{\Sigma}$ is a weighted average of $\Sigma_t$.

So we have that $\mathrm{dist}(\tilde{\Sigma}, \hat{\Sigma}) \leq \alpha/3$ and $\mathrm{dist}(\Sigma, \hat{\Sigma}) \leq \alpha$ w.p. $\geq 1 - \xi$. Applying the $3/2$-approximate triangle inequality for the dist function, we get:

$$\|\Sigma^{-1/2} \tilde{\Sigma} \Sigma^{-1/2} - I\| \leq \frac{3}{2}(\alpha + \frac{\alpha}{3}) = 2\alpha \qquad \square$$

**Lemma A.5.** (Privacy Analysis) Suppose that:

$$n \geq 800m \cdot \left( \max \left\{ \sqrt{\frac{2d(d + 1/\eta^2)}{\epsilon}}, \frac{8d\sqrt{\ln(2/\delta)}}{\epsilon}, \frac{8\ln(2/\delta)}{\epsilon}, \frac{12\sqrt{d\ln(2/\delta)}}{\epsilon\eta} \right\} \right)$$

Then Algorithm 3 is $(2\epsilon, 4e^\epsilon \delta)$-DP.

*Proof.* By basic composition of Truncated Laplace Mechanism and Lemma 3.6 of Ashtiani & Liaw (2022). $\square$

Both Lemma A.4 and Lemma A.5 together imply the following theorem:

**Theorem A.6.** *Let $\xi$, $\epsilon$, $\delta$ be parameters, and let $X \subseteq \mathbb{R}^d$ be a $(m, \alpha, \beta)$-subsamplable set of $n$ points. Then, for Algorithm 3, the following properties hold:*

1. *Algorithm 3 satisfies $(2\epsilon, 4e^\epsilon \delta)$-Differential Privacy.*

2. *The algorithm returns $\tilde{\Sigma}$ such that $dist(\Sigma, \tilde{\Sigma}) \leq 2\alpha$, where $\Sigma = \frac{1}{n} X X^\top$.*

*These guarantees hold under the following conditions:*

1. *The dataset size satisfies:*

$$n \geq 800m \cdot \left( \max \left\{ \sqrt{\frac{2d(d + 1/\eta^2)}{\epsilon}}, \frac{8d\sqrt{\ln(2/\delta)}}{\epsilon}, \frac{8\ln(2/\delta)}{\epsilon}, \frac{12\sqrt{d\ln(2/\delta)}}{\epsilon\eta}, \frac{\ln\left(1 + \frac{e^\epsilon - 1}{2\delta}\right)}{80\epsilon} \right\} \right).$$

   *where $\eta = \frac{\alpha}{48C(\sqrt{d} + \sqrt{\ln(4/\xi)})}$ for a sufficiently large constant $C > 0$.*

2. *The subsamplability parameters satisfy $m \geq \frac{2\beta n}{\xi}$.*

# B   Finding Initial Parameters

Our recursive algorithm requires as input two parameters that characterize the "aspect ratio" of the input, namely  – $R_{\max}$, the maximum distance of any point from the origin, and $\lambda_{\min}$, the minimum eigenvalue of the input. These two parameters give us the initial bounds, as they imply that $\lambda_{\min} I \preceq \Sigma \preceq R_{\max}^2 I$. We detail here the algorithms that allow us to retrieve these parameters. Finding $R_{\max}$ is fairly simple, as it requires us only to apply off-the-shelf algorithms that find an enclosing ball of the $n$ input points (Nissim et al. (2016); Nissim & Stemmer (2018); Mahpud & Sheffet (2022)). Finding the minimal eigenvalue is fairly simple as well, and we use a subsample-and-aggregate framework. Details follow.

**Finding a Covering Radius of The Data**

**Definition B.1.** A *1-cluster problem* $(X^d, n, t)$ consists of a $d$-dimensional domain $X^d$ and parameters $n \geq t$. We say that algorithm $\mathcal{M}$ solves $(X^d, n, t)$ with parameters $(\Delta, \omega, \beta)$ if for every input database $S \in (X^d)^n$ it outputs, with probability at least $1 - \beta$, a center $c$ and a radius $r$ such that: (i) the ball of radius $r$ around $c$ contains at least $t - \Delta$ points from $S$; and (ii) $r \leq w \cdot r_{\mathrm{opt}}$, where $r_{\mathrm{opt}}$ is the radius of the smallest ball in $X^d$ containing at least $t$ points from $S$.

**Theorem B.2** (Nissim & Stemmer (2018)). *Let $n, t, \beta, \epsilon, \delta$ be s.t.*

$$t \geq O\left(\frac{n^{0.1} \cdot \sqrt{d}}{\epsilon} \log\left(\frac{1}{\beta}\right) \log\left(\frac{nd}{\beta\delta}\right) \sqrt{\log\left(\frac{1}{\beta\delta}\right)} \cdot 9^{\log^*(2|X|\sqrt{d})}\right)$$

*There exists an $(\epsilon, \delta)$-differentially private algorithm that solves the 1-cluster problem $X^d, n, t$ with parameters $(\Delta, \omega)$ and error probability $\beta$, where $\omega = O(1)$ and*

$$\Delta = O\left(\frac{n^{0.1}}{\epsilon} \log\left(\frac{1}{\beta\delta}\right) \log\left(\frac{1}{\beta}\right) \cdot 9^{\log^*(2|X|\sqrt{d})}\right)$$

In words, there exists an efficient $(\epsilon, \delta)$-differentially private algorithm that (ignoring logarithmic factors) is capable of identifying a ball of radius $O(r_{opt})$ containing $t - \tilde{O}(\frac{n^{0.1}}{\epsilon})$ points, provided that $t \geq \tilde{O}(n^{0.1} \cdot \sqrt{d}/\epsilon)$.

**Finding Minimal Eigenvalue** The algorithm described in Kamath et al. (2022) (Section 3) privately estimates all eigenvalues of the second moment matrix of the data. However, for the purpose of this study, we focus solely on identifying the minimum eigenvalue while maintaining the privacy guarantees provided by the algorithm. To adapt the algorithm, we modify its structure to prioritize the computation of the minimum eigenvalue directly, rather than estimating the full spectrum of eigenvalues. This simplification not only reduces computational overhead but also aligns with the specific objectives of our work. Below, we detail the adjusted methodology and highlight the changes made to the original theorem.

---

**Algorithm 4** DP Minimum Eigenvalue Estimator

---

**Input:** a set of $n$ points $X \subseteq \mathbb{R}^d$, subsamplability parameters $m, \alpha, \beta$, error parameter $\xi \in (0, 1)$, privacy parameters $\epsilon, \delta$.

1: Randomly split $X$ into $T = \lfloor n/m \rfloor$ subgroups $X_1, \ldots, X_T$ of size $m$.
2: **for** $t \in [T]$ **do**
3:     Let $\lambda_{min}^t$ be the minimum eigenvalue of $\frac{1}{m} X_t X_t^T$.
4: **end for**
5: $\Omega \leftarrow \{\ldots, [(1-\alpha)^2, 1-\alpha), [1-\alpha, 1), [1, \frac{1}{1-\alpha}), [\frac{1}{1-\alpha}, \frac{1}{(1-\alpha)^2}), \ldots\} \cup \{[0, 0]\}$.
6: Divide $[0, \infty)$ into $\Omega$.
7: Run $(\epsilon, \delta)$-DP histogram on all $\lambda_{min}^t$ over $\Omega$.
8: **if** no bucket is returned **then**
9:     **return** $\perp$.
10: **end if**
11: Let $[\ell, u]$ be a non-empty bucket returned and set $\tilde{\lambda}_{min} \leftarrow \ell$.
12: **return** $\tilde{\lambda}_{min}$

---

**Theorem B.3.** *((Differentially Private EigenvalueEstimator) from Kamath et al. (2022)) For every $\epsilon, \delta, \xi > 0$, the following properties hold for Algorithm 4:*

*1. The algorithm is $(\epsilon, \delta)$-differentially private.*

*2. The algorithm runs in time $poly(n/m, \ln(1/\epsilon\xi))$*

*3. if:*

$$n \geq O\left(\frac{m \ln(1/\delta\xi)}{\epsilon}\right) \quad and \quad m \geq \frac{2\beta n}{\ln\left(\frac{1}{1-\xi/2}\right)}$$

*then it outputs $\tilde{\lambda}_{min}$ such that with probability at least $1 - \xi$, $\tilde{\lambda}_{min} \in [(1-\alpha)\lambda_{min}, (1+\alpha)\lambda_{min}]$ where $\lambda_{min}$ is the minimum eigenvalue of $\frac{1}{n} X X^T$.*

*Proof.* Privacy and running time is proven by the theorem of stability-based private histograms (See Lemma 2.6 in Kamath et al. (2022)). Now, we move on to the accuracy guarantees. By subsamplability, with probability at least $1 - \xi/2$, the non-private estimates of $\lambda_{min}$ must be within a factor of $(1 \pm \alpha)$ due to our subsample complexity. Therefore, at most two consecutive buckets would be filled with $\lambda_{min}^t$s. Due to our sample complexity and private histograms utility, those buckets are released with probability at least $(1 - \xi/2)$, which proves our theorem. □

## C  Missing Proofs

**Lemma C.1.** (Lemma 3.4 restated.) Given $X = \{X_1, ..., X_n\} \subset \mathbb{R}^d$, $C > 0, c > 0, 0 < \eta < 1, 0 < \psi < 1$, and $\kappa \geq 1$ s.t. $I \preceq \Sigma \preceq \kappa I$ where $\Sigma = \frac{1}{n} X X^T$. Let $V \leftarrow \text{Span}(\{v_i : \lambda_i \geq \psi\kappa)$ and $\Pi \leftarrow \eta\Pi_V + \Pi_{V^\perp}$. Given that: $n \geq O\left(m\sqrt{\frac{dT}{\rho}}\ln(T/\xi)\right)$ Then:

$$(1 - \frac{1}{\eta^2\psi - c} \cdot \frac{1}{\kappa})I \preceq \Pi\Sigma\Pi \preceq (\eta^2 + \psi + 2c)\kappa I$$

In particular, if $\kappa > C$ for some constant $C$ then:

$$I \preceq \frac{1}{(1 - \frac{1}{\eta^2\psi - c})}\Pi\Sigma\Pi \preceq \frac{\eta^2 + \psi + 2c}{(1 - \frac{1}{C(\eta^2\psi - c)})}\kappa I$$

*Proof.* First we prove the upper bound. Note that $\|\Pi\Sigma\Pi\|_2 = \|\Pi(\tilde{\Sigma} - N)\Pi\| \leq \|\Pi\tilde{\Sigma}\Pi\| + \|N\|$. So we bound the two terms separately. Using Fact 2.1 with $A = O(\frac{1}{d}\ln(1/\xi))$ for sufficiently large $n$ we get $\|N\|_2 \leq c\kappa$ w.p. $1 - \xi/2$. Additionally $\|\Pi\tilde{\Sigma}\Pi\|_2 \leq \eta^2\|\Pi_V\tilde{\Sigma}\Pi_V\| + \|\Pi_{V^\perp}\tilde{\Sigma}\Pi_{V^\perp}\| \leq \eta^2(\kappa + c) + \psi\kappa \leq (\eta^2 + \psi + c)\kappa$. So overall,

$$\|\Pi\Sigma\Pi\|_2 \leq (\eta^2 + \psi + 2c)\kappa$$

Next we prove the lower bound. Let $u \in \mathcal{S}^{d-1}$. Our lower bound requires we show that $u^T\Pi\Sigma\Pi u \geq (1 - \frac{1}{\eta^2\psi - c} \cdot \frac{1}{\kappa})$. We consider two cases:

- Case 1:  $\|\Pi_V u\|^2 < \frac{1}{\eta^2\psi - c} \cdot \frac{1}{\kappa}$.
  Since $\|\Pi_{V^\perp}u\|^2 + \|\Pi_V u\|^2 = 1$ we have $\|\Pi_{V^\perp}u\|^2 > (1 - \frac{1}{\eta^2\psi - c} \cdot \frac{1}{\kappa})$, hence, using the fact that $\Sigma \succeq I$ we have that

$$u^T\Pi\Sigma\Pi u \geq \|\Pi u\|^2 \geq \|\Pi_{V^\perp}u\|^2 > (1 - \frac{1}{\eta^2\psi - c} \cdot \frac{1}{\kappa})$$

- Case 2:  $\|\Pi_V u\|^2 \geq \frac{1}{\eta^2\psi - c} \cdot \frac{1}{\kappa}$.
  Note that $u^T\Pi\Sigma\Pi u = u^T\Pi(\tilde{\Sigma} - N)\Pi u = u^T\Pi\tilde{\Sigma}\Pi u - u^T\Pi N\Pi u$. So we bound each term separately. We know that

$$u^T\Pi\tilde{\Sigma}\Pi u \geq u^T\Pi_V\tilde{\Sigma}\Pi_V u \geq \eta^2\psi\kappa\|\Pi_V u\|^2$$

  Additionally, based on the bound on the spectral norm of $N$ we have that   $u^T\Pi N\Pi u \leq c\kappa\|\Pi u\|^2 \leq c\kappa\|\Pi_V u\|^2$. So overall:

$$u^T\Pi\Sigma\Pi u \geq (\eta^2\psi - c)\kappa\|\Pi_V u\|^2 \geq 1 \qquad \square$$

**Claim C.2.** (Claim 3.7 restated.) Analogously to Definition 3.2, fix any $m' \geq m$ and define $P_{\text{tail}}(m') = \{x \in X : \exists u \in \mathbb{R}^d : (x^T u)^2 > m'(1 + \alpha)\frac{1}{n}\sum_i(x_i^T u)^2\}$. Then it holds that

$$\Pr_{x \in_R X}[x \in P_{\text{tail}}(m')] \leq \frac{\beta + \beta^2}{m'}.$$

*Proof.* Let $x \in X$ be a datapoint and $\hat{X} = \{\hat{X}_1, \ldots, \hat{X}_{m'}\}$ be a subsample of $m'$ points i.i.d from $X$. From subsamplability we know that: $\Pr[(1-\alpha)\Sigma \preceq \hat{\Sigma} \preceq (1+\alpha)\Sigma] \geq 1 - \beta$ where $\Sigma = \frac{1}{n}\sum_{x \in X} xx^T$ and $\hat{\Sigma} = \frac{1}{m'}\sum_{i \in [m']} \hat{X}_i\hat{X}_i^T$. In other words:

$$\Pr[\forall u \in \mathbb{R}^d : (1 - \alpha)u^T\Sigma u \leq u^T\hat{\Sigma}u \leq (1 + \alpha)u^T\Sigma u] \geq 1 - \beta.$$

Clearly, if even a single $x_0 \in P_{\text{tail}}(m')$ belongs to $\hat{X}$ then we'd have that for some direction $u$ we have

$$u^T\hat{\Sigma}u = \frac{1}{m'}\sum_i(\hat{X}_i^T u)^2 \geq \frac{1}{m'}(x_0^T u)^2 > (1 + \alpha)u^T\Sigma u$$

contradicting subsamplability. It follows that w.p. $\geq 1 - \beta$ no point in $\hat{X}$ belongs to $P_{\text{tail}}(m')$. Thus, if we denote $p = \Pr_{x \in_R X}[x \in P_{\text{tail}}(m')]$ then we get that $1 - \beta \leq (1-p)^{m'}$. Using known inequalities we get $e^{-pm'} \geq (1-p)^{m'} \geq 1 - \beta \geq e^{-\beta - \beta^2}$ therefore $p \leq \frac{\beta + \beta^2}{m'}$. $\qquad\square$

**Lemma C.3.** (Lemma 3.8 restated.)    Let $X$ be a $(m, \alpha, \beta)$-subsamplable with $\beta \leq \frac{\alpha}{4(1+\alpha)\log(\frac{R^2}{(1+\alpha)m})}$. Let $P = X \setminus P_{\text{tail}}$. Then:

$$\forall u \in \mathbb{R}^d : \ \frac{1}{n} \sum_{x \in P} \langle x, u \rangle^2 \geq (1 - \alpha) \frac{1}{n} \sum_{x \in X} \langle x, u \rangle^2$$

*Proof.* Fix direction $u$, and denote $\lambda^P = \frac{1}{n} \sum_{x \in P} \langle x, u \rangle^2$ and $\lambda = \frac{1}{n} \sum_{x \in X} \langle x, u \rangle^2$. Our goal is to prove that $\lambda - \lambda^P \leq \alpha\lambda$.

Assume that $P_{\text{tail}}$ holds a $p$-fraction of $X$, and denote also $\lambda^{\text{tail}} = \frac{1}{|P_{\text{tail}}|} \sum_{x \in P_{\text{tail}}} \langle x, u \rangle^2$. Hence:

$$\lambda = p\lambda^{\text{tail}} + \lambda^P \implies \lambda - \lambda^P = p\lambda^{\text{tail}}$$

which implies that our goal is to prove that $p\lambda^{\text{tail}} \leq \alpha\lambda$.

Now split the interval $[(1 + \alpha)m\lambda, R^2]$, the interval of $P_{\text{tail}}$, into buckets $B_0, B_1, B_2, ..., B_k$ where

$$B_i = \left[2^i(1 + \alpha)m\lambda, 2^{i+1}(1 + \alpha)m\lambda\right)$$

and $k = \log(\frac{R^2}{(1+\alpha)m\lambda}) \leq \log(\frac{R^2}{(1+\alpha)m})$, and clearly we have $\Pr_{x \in X}[\langle x, u \rangle^2 > m(1+\alpha)\lambda] = \Pr_{x \in X}[x \in \bigcup_{i=0}^{k} B_i]$. Recall that Claim 3.7 implies that for any $i$ we can set $m' = 2^i m$ and get:

$$\Pr_{x \in X}[x \in B_i] \leq \frac{\beta + \beta^2}{2^i m}$$

Now we can bound $p\lambda^{P_{\text{tail}}}$ using $B_i$:

$$p\lambda^{\text{tail}} \leq \sum_{i=0}^{k} \Pr_{x \in X}[x \in B_i] \cdot 2^{i+1}(1 + \alpha)m\lambda$$

$$\leq \sum_{i=0}^{k} \frac{\beta + \beta^2}{2^i m} \cdot 2^{i+1}(1 + \alpha)m\lambda \leq 2\beta \cdot 2k(1 + \alpha)\lambda$$

which is upper bounded by $\lambda$ for $\beta \leq \frac{\alpha}{4(1+\alpha)k}$. $\qquad\square$

**Lemma C.4.** (Lemma 3.9 restated.) At each iteration $t$ of Algorithm 2, only points belonging to $P_{\text{tail}}$ are subjected to shrinking, given that $\alpha < 1/2$, $\psi = \frac{1}{10m}$ and $c = \frac{1}{80m}$.

*Proof.* The proof works by induction on the iterations of the algorithm. Clearly, at $t = 0$, before the algorithm begins, no points were subjected to shrinking so the argument is vacuously correct.

Let $t \leq T$ and let $x \in X^t$ denote a point that undergoes shrinking *for the first time* at iteration $t$, with $X^t$ denoting the input of the $t$-th time we apply Algorithm 2. Our goal is to show that $x$ stems from a point in $P_{\text{tail}}$. Since $x$ hasn't been shrunk prior to iteration $t$, then there exists some $x_i$ in the original input such that $x = \Lambda^t x_i$ for the linear transformation $\Lambda^t = \Pi^t \cdot \Pi^{t-1} \cdot ... \cdot \Pi^1$. Our goal is to show that $x_i \in P_{\text{tail}}$.

We assume $x$ is shrunk at iteration $t$. This shrinking happens since $x^T \Pi_{V^\perp} x > \frac{3}{7}R^2$. With $\alpha \leq \frac{1}{2}$ we have $\frac{3}{7} \geq \frac{1+\alpha}{4}$, then it follows that $x$ satisfies:

$$x^T \Pi_{V^\perp} x > \frac{1}{4}(1 + \alpha)R^2 = \frac{1}{4}(1 + \alpha)\kappa,$$

For the given parameters $\psi$ and $c$, observe that:

$$x^T \Pi_{V^\perp} x > \frac{1}{4}(1 + \alpha)\kappa > 2m(1 + \alpha)(\psi + c)\kappa + c\kappa.$$

Recall that $V^\perp$ denotes the subspace spanned by all eigenvectors of $\tilde{\Sigma} = \Sigma + N$ corresponding to eigenvalues $\leq \psi\kappa$ and that $\|N\| \leq c\kappa$. Now denote $U^\perp$ the subspace spanned by all eigenvectors of $\Sigma$ corresponding to eigenvalues $\leq (\psi + c)\kappa$. By Weyl's theorem it holds that

$$x^T \Pi_{U^\perp} x \geq x^T \Pi_{V^\perp} x - c\kappa \geq 2m(1+\alpha)(\psi+c)\kappa$$

Thus we infer the existence of $u \in U^\perp$ (unit length vector in the direction $\Pi_{U^\perp} x$) such that:

$$(x^T u)^2 > 2m(1+\alpha)(\psi+c)\kappa.$$

But as $U^\perp$ is spanned by all eigenvalues $\leq (\psi+c)\kappa$ of $\Sigma$ then it holds that $\frac{1}{n}\sum_{x\in X^t}(x^T u)^2 \leq (\psi+c)\kappa$, so

$$(x^T u)^2 > 2m(1+\alpha) \cdot \frac{1}{n}\sum_{x\in X^t}(x^T u)^2 \tag{2}$$

Now recall that $x$ undergoes shrinking for the first time at iteration $t$. That means that there exists some $x_i$ in the original input such that $x = \Lambda^t x_i$ for the linear transformation $\Lambda^t = \Pi^t \cdot \Pi^{t-1} \cdot \ldots \cdot \Pi^1$. Moreover, by the induction hypothesis all points that were shrunk upto iteration $t$ are from $P_{\text{tail}}$. So for any point $z \in X^t \setminus P_{\text{tail}}$ it holds that $z = \Lambda^t y$ for the corresponding $y$ in the original input $X$. We get that $\frac{1}{n}\sum_{x\in X^t}(x^T u)^2 \geq \frac{1}{n}\sum_{x\in X^t\setminus P_{\text{tail}}}(x^T u)^2 = \frac{1}{n}\sum_{y\in X\setminus P_{\text{tail}}}(y^T \Lambda^T u)^2$.

We now apply Lemma 3.8 to infer that

$$\frac{1}{n}\sum_{y\in X\setminus P_{\text{tail}}}(y^T\Lambda^T u)^2 \geq (1-\alpha)\frac{1}{n}\sum_{y\in X}(y^T\Lambda^T u)^2 \geq \frac{1}{2n}\sum_{y\in X}(y^T\Lambda^T u)^2$$

seeing as $\alpha \leq 1/2$. Plugging this into Equation (2) we

$$\langle x_i, \Lambda^T u\rangle^2 = (x^T u)^2 > m(1+\alpha) \cdot \frac{1}{n}\sum_{y\in X}\langle y, \Lambda^T u\rangle^2$$

which by definition proves that $x_i$ belongs to $P_{\text{tail}}$. □

**Corollary C.5.** (Corollary 3.10 restated.) In all iterations of the algorithm it holds that $\Sigma \succeq I$, namely, that the least eigenvalue of the second moment matrix of the input is $\geq 1$.

*Proof.* Again, we prove this by induction of $t$, the iteration of Algorithm 2. In fact, denoting $X^t$ as the input of of Algorithm 2 at iteration $t$, then we argue that the least eigenvalue of the matrix $\frac{1}{n}\sum_{x\in X^t\setminus P_{\text{tail}}} xx^T$ is at least 1.

Consider $t = 0$, prior to the execution of Algorithm 2 even once. Apply Lemma 3.8 with $u$ being the direction of the least eigenvalue of $\Sigma$, and we get that $\frac{1}{n}\sum_{x\in X\setminus P_{\text{tail}}} xx^T \geq (1-\alpha) \cdot 1$. Observe that Algorithm 1 invokes Algorithm 2 on the input multiplied a $\frac{1}{1-\alpha}$-factor, and so it holds that $\frac{1}{n}\sum_{x\in X^0\setminus P_{\text{tail}}} xx^T \geq 1$ for $X^0$, the very first input on which Algorithm 2 is run.

Consider now any intermediate $t$, where we assume that input satisfies $\frac{1}{n}\sum_{x\in X^t\setminus P_{\text{tail}}} xx^T \geq 1$. We can apply Lemma 3.4 and Corollary 3.5 solely to the points in $X^t \setminus P_{\text{tail}}$ and have that $\frac{1}{n}\sum_{x\in X^t\setminus P_{\text{tail}}} \frac{8}{7}x\Pi x^T \geq 1$. Since Lemma 3.9 asserts no point in $X^t \setminus P_{\text{tail}}$ is shrunk then we get that the required also holds at the invocation of the next iteration. □

**Claim C.6.** (Claim 4.2 restated.) Let $\mathcal{D}$ be a distribution on $\mathbb{R}^d$ with a finite second moment $\Sigma$. Consider a random vector $y$ chosen by drawing $x \sim D$ and then multiplying $y = \Sigma^{-1/2}x$. Suppose that $\|y\| \leq M_1$ a.s. that we also have a bound $\|\mathbb{E}[(y^T y)yy^T]\| \leq M_2$. Fix $\alpha, \beta > 0$. If we draw $m = \max\left\{\frac{2M_2}{\alpha^2}, \frac{2(1+M_1^2)}{3\alpha}\right\} \cdot \ln(4d/\beta)$ examples from $\mathcal{D}$ and compute the empirical second moment matrix $\hat{\Sigma}$, then w.p. $\geq 1 - \beta$ it holds that

$$\|\Sigma^{-1/2}\hat{\Sigma}\Sigma^{-1/2} - I\|_2 \leq \alpha$$

*Proof.* Denote our sample of drawn points as $x_1, x_2, ..., x_m$. Define $\forall i : y_i = \Sigma^{-1/2}x_i$ so that $\mathbb{E}[y_i y_i^T] = I$. This transforms the problem to bounding: $\|\hat{\Sigma}_y - I\|_2 \leq \alpha$ where $\hat{\Sigma}_y = \frac{1}{m}\sum_{i=1}^{m} y_i y_i^T$.

Now $\|y_i y_i^T\|_2 = \|y_i\|_2^2 \leq M_1^2$.

Next, define the random deviation $Z$ of the estimator $\hat{\Sigma}_y$ from the true covariance matrix $I$:

$$Z = \hat{\Sigma}_y - I = \sum_{i=1}^{m} Z_i \quad \text{where } Z_i = \frac{1}{m}(y_i y_i^T - I)$$

The random matrices $Z_i$ are independent, identically distributed, and centered. To apply Fact 4.1, we need to find a uniform bound $R$ for the summands, and we need to control the matrix variance statistic $\sigma^2$. First, let us develop a uniform bound on the spectral norm of each summand. We can calculate that:

$$\|Z_i\| = \frac{1}{m}\|y_i y_i^T - I\| \leq \frac{1}{m}(\|y_i y_i^T\| + \|I\|) \leq \frac{M_1 + 1}{m}$$

Second, we need to bound the matrix variance statistic $\sigma^2$ defined in 4.1, with $\sigma^2 = \|\mathbb{E}[ZZ^T]\| = \|\sum_{i=1}^{m} \mathbb{E}[Z_i Z_i^T]\|$. We need to determine the variance of each summand. By direct calculation:

$$\mathbb{E}[Z_i Z_i^T] = \frac{1}{m^2}\mathbb{E}[(y_i y_i^T - I)(y_i y_i^T - I)^T] = \frac{1}{m^2}(\mathbb{E}[y_i y_i^T y_i y_i^T] - I) \preceq \frac{1}{m^2}\mathbb{E}[y_i y_i^T y_i y_i^T]$$

Then we have:

$$\sigma^2 = \|\sum_{i=1}^{m} \mathbb{E}[Z_i Z_i^T]\| \leq \|\sum_{i=1}^{m} \frac{1}{m^2}\mathbb{E}[y_i y_i^T y_i y_i^T]\| \leq \frac{1}{m}\|\mathbb{E}[(y_i^T y_i)y_i y_i^T]\| \leq \frac{M_2}{m}$$

We now invoke the matrix Bernstein inequality, Fact 4.1:

$$\Pr[\|\hat{\Sigma}_y - I\|_2 > \alpha] \leq 2d \cdot \exp\left(-\frac{m\alpha^2/2}{M_2 + \alpha(1 + M_1^2)/3}\right)$$

which is upper bounded by $\beta$ given that $m \geq m(\alpha, \beta) = \max\left\{\frac{2M_2}{\alpha^2}, \frac{2(1+M_1^2)}{3\alpha}\right\} \cdot \ln(4d/\beta)$. $\qquad\square$

# D   More Applications

## D.1   Application: The Uniform Distribution Over Some Convex Ellipsoid

Fix a PSD matrix $0 \preceq A \preceq I$. Consider the uniform distribution $\mathcal{D}$ over the surface of some convex ellipsoid $\mathcal{K} = \{x \in \mathbb{R}^d \mid x^T A^{-1} x = 1\}$. Our goal in this section is to argue that our algorithm is able to approximate the 2nd moment matrix $\Sigma_{\mathcal{D}}$. To that end, we want to determine the size $m$ of a subsample drawn from $\mathcal{D}$, such that with probability at least $1 - \beta$: $\|\Sigma_{\mathcal{D}}^{-1/2}\hat{\Sigma}_{\mathcal{D}}\Sigma_{\mathcal{D}}^{-1/2} - I\|_2 \leq \alpha$ where $\Sigma_{\mathcal{D}} = \frac{1}{d}A$ is the second moment of $\mathcal{D}$.

To utilize Claim 4.2, it is necessary to compute the bounds $M_1$ and $M_2$. First, note that if $x$ is drawn from the surface of $\mathcal{K}$ then $\|\Sigma_{\mathcal{D}}^{-1/2}x\| = \sqrt{d}\|y\|$ for unit-length vector $y$, implying $M_1 = \sqrt{d}$. Second, consider $y = \Sigma_{\mathcal{D}}^{-1/2}x$ and observe that:

$$\mathbb{E}[yy^T yy^T] = \mathbb{E}[\Sigma_{\mathcal{D}}^{-1/2}xx^T\Sigma_{\mathcal{D}}^{-1}xx^T\Sigma_{\mathcal{D}}^{-1/2}] = d^2 \cdot \mathbb{E}[A^{-1/2}xx^T A^{-1}xx^T A^{-1/2}]$$
$$\overset{(*)}{=} d^2 \cdot \mathbb{E}[A^{-1/2}xx^T A^{-1/2}] = d^2 \cdot A^{-1/2}\mathbb{E}[xx^T]A^{-1/2} \overset{(**)}{=} dI$$

where inequality $(*)$ follows from the fact that $x^T A^{-1}x = 1$ for all $i$ and equality $(**)$ follows since $\mathbb{E}[x_i x_i^T] = \frac{1}{d}A$.

Hence we have $\|\mathbb{E}[y_i y_i^T y_i y_i^T]\| \leq d = M_2$, and we conclude that $\mathcal{D}$ is $(O\left(\frac{d}{\alpha^2} \cdot \ln(4d/\beta)\right), \alpha, \beta)$-subsamplable.

Recall that we $(1 \pm \gamma)$-approximate the 2nd-moment matrix *of the input* w.p.$\geq 1 - \xi$. Thus, we need the input itself to be a $(1 \pm \gamma)$-approximation of the 2nd-moment matrix of the distribution. This means our algorithm requires

$$m(\gamma, \xi) + O\left(\frac{m(\alpha, \beta)}{\gamma}\sqrt{\frac{d}{\rho} \cdot \log(\frac{1}{\lambda_{\min}})} \log(\log(\frac{1}{\lambda_{\min}})/\xi)\right)$$

for $\alpha = 1/2$ and $\beta = O(\frac{1}{\log(1/\lambda_{\min})})$ in order to return a $(1 \pm O(\gamma))$-approximation of the 2nd moment of the distribution w.p. $\geq 1 - O(\xi)$.

Plugging the $m(\alpha, \beta)$ of $\mathcal{D}$ we conclude that in order to return a $(1 \pm O(\gamma))$-approximation of the 2nd moment of $\mathcal{D}$ w.p. $\geq 1 - O(\xi)$ our sample complexity ought to be

$$m(\gamma, \xi) + O\left(\frac{m(\frac{1}{2}, \frac{1}{\log(1/\lambda_{\min})})}{\gamma}\sqrt{\frac{d}{\rho} \cdot \log(1/\lambda_{\min})} \cdot \log(\frac{\log(1/\lambda_{\min})}{\xi})\right) = \tilde{O}\left(\frac{d}{\gamma^2} + \frac{d^{3/2}}{\gamma\sqrt{\rho}}\right)$$

While, for comparison, our baseline algorithm requires

$$\tilde{O}\left(\frac{d \cdot m(\gamma, \xi)}{\epsilon\gamma}\right) = \tilde{O}\left(\frac{d^2}{\epsilon\gamma^3}\right)$$

in order to return a $(1 \pm O(\gamma))$-approximation of the 2nd moment of the distribution w.p. $\geq 1 - O(\xi)$.

### D.2    Examples of Heavy-Tailed Distributions

The above discussion holds for a general distribution. Next we demonstrate our algorithms performance a few heavy-tailed distributions. However, we also emphasize that many more applications are possible, since the subsamplability assumption is broad and encompasses a wide range of input distributions. For example, we further analyze datasets drawn from uniform distributions over ellipsoids and from Gaussian mixtures with stochastic outliers in Appendix D.1 and **??** respectively.

**The Truncated Pareto Distribution.**    Throughout we use the following distribution truncated Pareto distribution, denoted $\mathcal{P}_6$, that is supported on the interval $[1, B]$ for some $B > 1$ (say $B = 10$), and whose PDF is $\propto x^{-6}$. Formally, its PDF is

$$f(x) = \begin{cases} \frac{5B^5}{B^5-1}x^{-6} & \text{if } 1 \leq x \leq B \\ 0 & \text{if } x < 1 \text{ or } x > B \end{cases}$$

so that $f$ integrates to 1. Simple calculations show that $\mu_6 \stackrel{\text{def}}{=} \mathbb{E}_{\lambda \sim \mathcal{P}_6}[\lambda] = \frac{5}{4} \cdot \frac{B(B^4-1)}{B^5-1}$, that $\sigma_6^2 \stackrel{\text{def}}{=}$ $\mathbb{E}_{\lambda \sim \mathcal{P}_6}[\lambda^2] = \frac{5}{3} \cdot \frac{B^2(B^3-1)}{B^5-1}$ and that $\mathbb{E}_{\lambda \sim \mathcal{P}_6}[\lambda^4] = 5 \cdot \frac{B^4(B-1)}{B^5-1}$.

We consider here two distributions composed of a $\lambda \sim \mathcal{P}_6$ and $v \in_R \mathcal{S}^{d-1}$. The first is $\lambda v$, namely a vector with direction distributed uniformly over the unit sphere and magnitude distributed according to the $\mathcal{P}_6$ distribution; and the second is $\lambda \circ v$, namely a $(d+1)$-dimensional vector with first coordinate drawn from $\mathcal{P}_6$, concatenated with a uniformly chosen vector from the unit sphere on the remaining $d$ coordinates.

**The $\lambda v$ Distribution.**    Define the random variable $x = \lambda v$, where $v$ is uniformly distributed from the unit sphere, and $\lambda \sim \mathcal{P}_6$. Let $\mathcal{D}$ be the distribution of $x$. Our goal is to argue that our algorithm is able to approximate the 2nd moment matrix $\Sigma_\mathcal{D}$ and outperforms the baseline(s). To that end, we want to determine the size $m(\alpha, \beta)$ of a subsample drawn from $\mathcal{D}$ so that with probability at least $1 - \beta$: $\|\Sigma_\mathcal{D}^{-1/2}\hat{\Sigma}_\mathcal{D}\Sigma_\mathcal{D}^{-1/2} - I\|_2 \leq \alpha$.

To utilize Claim 4.2, we compute $\Sigma_\mathcal{D} = \mathbb{E}[\lambda v(\lambda v)^T] = \mathbb{E}[\lambda^2]\mathbb{E}[vv^T] = \frac{\sigma_6^2}{d}I$, since $\mathbb{E}[vv^T] = \frac{1}{d}I$. It follows that $y = \Sigma_\mathcal{D}^{-1/2}x = \frac{\sqrt{d}}{\sigma_6}\lambda v$, so we can bound $\|y\| = M_1 \stackrel{\text{def}}{=} \frac{B\sqrt{d}}{\sigma_6}$. Lastly, it is necessary to compute $M_2$. To this end, we evaluate the expectation:

$$\mathbb{E}[\Sigma_\mathcal{D}^{-1/2}x_ix_i^T\Sigma_\mathcal{D}^{-1}x_ix_i^T\Sigma_\mathcal{D}^{-1/2}] = \mathbb{E}[(\frac{B^5-1}{B^2(B^3-1)})^2 \cdot (\frac{3d}{5})^2 \cdot \lambda v(\lambda v)^T \cdot \lambda v(\lambda v)^T]$$

$$= \frac{9d^2}{25}\left(\frac{B^5-1}{B^2(B^3-1)}\right)^2 \cdot \mathbb{E}[\lambda^4] \cdot \mathbb{E}[vv^T vv^T] \overset{(*)}{=} \frac{9d^2}{25}\left(\frac{B^5-1}{B^2(B^3-1)}\right)^2 \cdot \mathbb{E}[\lambda^4] \cdot \mathbb{E}[vv^T]$$

$$\overset{(**)}{=} \frac{9d^2}{25}\left(\frac{B^5-1}{B^2(B^3-1)}\right)^2 \cdot \frac{5B^4(B-1)}{B^5-1} \cdot \frac{1}{d}I \leq 2dI$$

where $(*)$ follows from $v^T v = 1$ and $(**)$ holds since $\mathbb{E}[\lambda^4] = \frac{5B^4(B-1)}{B^5-1}$ and $\mathbb{E}[vv^T] = \frac{1}{d}I$. Hence we have $\|\mathbb{E}[y_i y_i^T y_i y_i^T]\| \leq 2d = M_2$. We now plug-in our values of $B$, $\lambda_{\min}$ and $M$ and infer this distribution is $m(\alpha,\beta)$-sampleable for $m = O(\max\{\frac{d}{\alpha^2}, \frac{dB^2}{\alpha}\}\ln(d/\beta))$. Plugging this into (1) we conclude that in order to return a $(1\pm O(\gamma))$-approximation of the 2nd moment of $\mathcal{D}$ w.p. $\geq 1-O(\xi)$ our sample complexity ought to be

$$m(\gamma,\xi)+O\left(\frac{m(\frac{1}{2},\frac{1}{\log(Bd)})}{\gamma}\sqrt{\frac{d}{\rho}\cdot\log(Bd)}\cdot\log(\frac{\log(Bd)}{\xi})\right) = \tilde{O}\left(\max\{\frac{d}{\gamma^2},\frac{dB^2}{\gamma}\} + \frac{dB^2}{\gamma}\sqrt{\frac{d}{\rho}}\right)$$

While, for comparison, our baseline algorithm requires

$$\tilde{O}\left(\frac{d\cdot m(\gamma,\xi)}{\epsilon\gamma}\right) = \tilde{O}\left(\frac{d}{\epsilon}\max\{\frac{d}{\gamma^2},\frac{B^2 d}{\gamma}\}\right)$$

in order to return a $(1\pm O(\gamma))$-approximation of the 2nd moment of the distribution w.p. $\geq 1-O(\xi)$.

**The $\lambda \circ v$ Distribution.** Consider the $(d+1)$-dimensional distribution $\mathcal{D}$ where the first coordinate is drawn from the truncated Pareto distribution $\mathcal{P}_6$ and the remaining coordinates are drawn uniformly over the unit sphere $\mathcal{S}^{d-1}$: I.e. $x = \begin{bmatrix}\lambda\\v\end{bmatrix} \sim \mathcal{D}$. Our goal in this section is to argue that our algorithm is able to approximate the 2nd moment matrix $\Sigma_\mathcal{D}$. To that end, we want to determine the size $m$ of a subsample drawn from $\mathcal{D}$, such that with probability at least $1-\beta$: $\|\Sigma_\mathcal{D}^{-1/2}\hat{\Sigma}_\mathcal{D}\Sigma_\mathcal{D}^{-1/2} - I\|_2 \leq \alpha$.

First it is easy to see that the $L_2$-norm of any $x \sim D$ is at most $B+1$. Next, we compute $\Sigma_\mathcal{D}$:

$$\Sigma_\mathcal{D} = \mathbb{E}\left[\begin{bmatrix}\lambda\\u\end{bmatrix}\begin{bmatrix}\lambda & u^T\end{bmatrix}\right] = \mathbb{E}\left[\begin{bmatrix}\lambda^2 & \lambda u^T\\\lambda u & uu^T\end{bmatrix}\right] = \begin{bmatrix}\sigma_6^2 & \bar{0}^T\\\bar{0} & \frac{1}{d}I\end{bmatrix}$$

It follows that the vector $y = \Sigma_\mathcal{D}^{-1/2}x = \begin{bmatrix}\sigma_6^{-1} & \bar{0}^T\\\bar{0} & \sqrt{d}I\end{bmatrix}\begin{bmatrix}\lambda\\u\end{bmatrix} = \begin{bmatrix}\sigma_6^{-1}\lambda\\\sqrt{d}u\end{bmatrix}$, so its norm is upper bounded by $M_1 = \sqrt{d+\sigma_6^{-2}B^2}$. So now we can evaluate the expectation:

$$\mathbb{E}[yy^T yy^T] = \mathbb{E}[\|y\|^2 yy^T] = \mathbb{E}\left[(\sigma_6^{-2}\lambda^2 + d)\begin{bmatrix}\sigma_6^{-2}\lambda^2 & \sigma_6^{-1}\sqrt{d}\lambda u^T\\\sigma_6^{-1}\sqrt{d}\lambda u & duu^T\end{bmatrix}\right]$$

$$= \begin{bmatrix}\sigma_6^{-4}\mathbb{E}[\lambda^4] + d\sigma_6^{-2}\mathbb{E}[\lambda^2] & 0\\0 & (\sigma_6^{-2}\mathbb{E}[\lambda^2]+d)I_d\end{bmatrix} = \begin{bmatrix}\sigma_6^{-4}\mathbb{E}[\lambda^4] + d & 0\\0 & (d+1)I_d\end{bmatrix}$$

$$= dI_{d+1} + \begin{bmatrix}\sigma_6^{-4}\mathbb{E}[\lambda^4] & 0\\0 & I_d\end{bmatrix}$$

seeing as $\sigma_6^{-4}\mathbb{E}[\lambda^4] \approx \frac{9}{5} = O(1)$ we can infer that $M_2 = \|\mathbb{E}[yy^T yy^T]\| = O(d)$.

We now plug-in our values of $B$, $\lambda_{\min}$ and $M$ and infer this distribution is $m(\alpha,\beta)$-samplable for $m = O(\max\{\frac{d}{\alpha^2}, \frac{d+B^2}{\alpha}\}\ln(d/\beta))$. Plugging this into (1) we conclude that in order to return a $(1 \pm O(\gamma))$-approximation of the 2nd moment of $\mathcal{D}$ w.p. $\geq 1 - O(\xi)$ our sample complexity ought to be

$$m(\gamma,\xi)+O\left(\frac{m(\frac{1}{2},\frac{1}{\log(Bd)})}{\gamma}\sqrt{\frac{d}{\rho}\cdot\log(Bd)}\cdot\log(\frac{\log(Bd)}{\xi})\right) = \tilde{O}\left(\max\{\frac{d}{\gamma^2},\frac{d+B^2}{\gamma}\} + \frac{d+B^2}{\gamma}\sqrt{\frac{d}{\rho}}\right)$$

The analysis suggests we can even set $B = O(\sqrt{d})$ and get a sample complexity of $\tilde{O}\left(\frac{d}{\gamma^2} + \frac{d^{3/2}}{\gamma\sqrt{\rho}}\right)$.

While, for comparison, our baseline algorithm requires

$$\tilde{O}\left(\frac{d \cdot m(\gamma, \xi)}{\epsilon\gamma}\right) = \tilde{O}\left(\frac{d}{\epsilon}\max\{\frac{d}{\gamma^3}, \frac{B^2 + d}{\gamma^2}\}\right) \overset{\text{if } B \leq \sqrt{d}}{=\!=} \tilde{O}(\frac{d^2}{\gamma^3\epsilon})$$

in order to return a $(1 \pm O(\gamma))$-approximation of the 2nd moment of the distribution w.p. $\geq 1 - O(\xi)$.

