# OpenReview forum: "A Private Approximation of the 2nd-Moment Matrix of Any Subsamplable Input"
_NeurIPS.cc/2025/Conference — NeurIPS 2025 poster_

### Official Review · Reviewer_GJ5K · 2025-06-28

**Clarity:** 3
**Significance:** 3
**Originality:** 3
**Rating:** 5
**Confidence:** 2

**Summary:**

Privately estimating statistics suchs as means and covariance matrices of a set of vectors has been a subject of intense study in the last 5-10 years. Good approximation is only possible under conditions on data, and various such conditions have been explored, often assuming that data is sampled from some "nice" distribution. This paper considers a new such condition that allows approximating the second moment of a distribution, namely subsamplability, which roughly means that a sample of the data has approximately the same spectral structure as the whole dataset with high probability. For datasets satisfying this condition, it is shown that a good approximation to the second moment matrix can be calculated with zero-concentrated differential privacy. This includes datasets with outliers, something not sufficiently captured by previous conditions.

**Questions:**

* Maybe a newbie question, but why don't you also consider covariance estimation? Is it because the results are weaker in that setting? Surely SOME result about covariance estimation must follow from your work

* Is there a private way to tell if the subsampling condition holds for given m, alpha and beta?

* In algorithm 2 the recursive call uses the same privacy parameter rho, which is counterintuitive since one would usually need to invoke composition. Can you comment on this?

* In Theorem 3.3 it sounds like large n is needed for privacy (property 1). If that is the case, how may one ensure that n is large enough for privacy?

**Ethical Concerns:**

["NO or VERY MINOR ethics concerns only"]

**Final Justification:**

The rebuttal confirms my assessment of the paper as a good theoretical contribution to the area.

**Limitations:**

The theoretical limitations are well discussed. I feel that the paper could have also discussed any limitations that may arise when trying to use this kind of method in practice.

**Quality:**

4

**Strengths And Weaknesses:**

The paper is a novel contribution to a well-studied area that is relevant to machine learning. Its relation to other results in the literature is well explained. Even though it is not in my area of expertise, I was able to understand the key ideas.

The work is theoretical with no empirical work, which can be seen as a weakness. The algorithm also needs to know the parameters for which the subsampling condition holds, and it is not obvious how to find such parameters in a private way.

---

> ### Author Rebuttal · Authors · 2025-07-30
>
> We thank the reviewers for their thoughtful feedback. We are pleased that our work is recognized as technically rigorous and novel, and we appreciate the opportunity to clarify and address the concerns raised.
>
> 1. On the subsamplability parameter: The parameter m is unknown in general, as it encapsulates a property of the input distribution. However, if the data is known to come from a specific distribution (e.g., heavy tailed distributions), the value of m can often be computed analytically — see Section 4.1 and Appendix D for such examples. In the general case, while there is no universally optimal way to estimate m under privacy, one can still privately run the algorithm for multiple candidate values of m, since our method is differentially private for any choice of m. This pragmatic approach allows post-hoc validation of outputs and circumvents the need for stringent concentration bounds.
> Note that even "classical" results, such as estimating under the promise of data sampled from a Gaussian, make assumptions on the nature of the input and do not concern themselves with verifying whether such an assumption holds or not. The only difference is that we use a slightly less common assumption. But, as it turns out, our subsamplability assumption makes it possible to deal with both cases at the same time: (1) data drawn from a distribution and (2) data with a small fraction (roughly \beta/m) of outliers. Lastly, we ourselves tried to address the "minimality" of subsamplability in lines 25-29 in the introduction.
>
> 2. On the lack of experiments: Our work is entirely theoretical, and all claims are rigorously proven. Empirical experiments in such settings merely illustrate that the mathematical analysis holds — a fact already established through our theorems. In this light, we believe that including synthetic experiments would not significantly enhance the contribution, although we agree that future applied extensions could and should incorporate them.
>
>
> Specific comments for Reviewer #3:
> 1. On covariance estimation:
> To estimate the covariance matrix, one typically subtracts the outer product of the mean vector. While this can be done privately using standard DP mean estimation algorithms, it incurs additional noise, hence degrading utility. For this rather technical reason, we focus on the second moment.
>
> 2. On privacy composition in recursion:
> Indeed, each recursive call uses the same target privacy parameter \rho, but we ensure that each call only consumes \rho/T-zCDP. By composition, the total privacy cost sums to \rho, preserving the global privacy budget.
>
> 3. On privacy and sample size:
> We appreciate this clarification. Privacy is preserved regardless of sample size. We agree the current wording in Theorem 3.3 may be misinterpreted, and we will revise it to: "Algorithm 1 preserves \rho-zCDP, and if n \geq ..."

---

> > ### Comment · Reviewer_GJ5K · 2025-08-05
> >
> > Thank you. The rebuttal confirms my assessment of the paper as a good theoretical contribution to the area.

---

> > > ### Author Response · Authors · 2025-08-06
> > > **Rating Disappeared**
> > >
> > > Thank you reviewer GJ5K, but for some reason we don't see your rating (5) anymore. Did you accidentally deleted it?

---

### Official Review · Reviewer_dJaK · 2025-06-30

**Clarity:** 2
**Significance:** 2
**Originality:** 3
**Rating:** 4
**Confidence:** 3

**Summary:**

This paper presents a differentially private algorithm for estimating the 2nd-moment matrix (covariance matrix) under the assumption of $(m,\alpha, \beta)$-subsamplability. The authors propose a recursive mechanism that iteratively reduces the range of the data while preserving spectral structure and satisfying zCDP. The proposed approach is robust to outliers and overcomes limitations in prior DP methods.

**Questions:**

Please see the weaknesses.

**Ethical Concerns:**

["NO or VERY MINOR ethics concerns only"]

**Final Justification:**

Nice theoretical work on the private estimation problem, although there are no numerical results.

**Limitations:**

Please see the weaknesses.

**Quality:**

3

**Strengths And Weaknesses:**

Strengths:

The proposed algorithm and its analysis are both technically rigorous and innovative.

Weaknesses:

1. The subsamplability assumption, although less stringent than standard matrix concentration bounds, remains somewhat probabilistic and non-constructive. It is unclear how one might verify or estimate whether a dataset satisfies this condition, especially under privacy constraints.
2. The paper lacks experimental results, which limits its practical validation. Simulations on synthetic or real-world datasets (e.g., robustness to outliers, improved sample complexity) and comparisons with author-mentioned methods such as Nissim et al.(2007), Kamath et al.  (2019), Brown et al. (2023) would strengthen the empirical relevance.
3. The recursive algorithm involves multiple spectral decompositions and tail-point truncation, which could be expensive in high dimensions.
4. It seems that the error bound analysis critically relies on the parameters $R$ and $\lambda_{\rm min}$, but the procedure for estimating these quantities in a privacy-preserving manner is only briefly mentioned. I am curious about how the $R$ and $\lambda_{\rm min}$ affect the main utility error bound.
5. In Section 4, how sensitive is the robustness guarantee to slight violations of the $\Sigma_{\rm out} \preceq O(1/\eta)\Sigma_{\rm D}$ condition in line 337?
6. In Algorithm 1, the iteration bound is defined as $T = \log_{7/3}\left(\frac{\frac{1}{1-\alpha}R^2}{640m}\right)$, which takes an unusual logarithmic base value $7/3$. Could the authors clarify the rationale behind choosing this specific base?
7. How should the covering radius $R$ be determined in practical scenarios where the dataset may contain outliers? Since the input may not be bounded.
8. The title of ``....\emph{Any} Subsamplable Input" may be overly broad, as the efficiency of the proposed method relies on the subsamplability assumption. Additionally, the algorithm uses uniform random subsampling. It would be useful to discuss whether the results extend to other subsampling schemes, such as Poisson or leverage-score based sampling.
9. The manuscript is not well-structured, and the content lacks logic. For example, the introduction presents subsamplability assumption in a way that is difficult to follow without prior knowledge. Further, Sections 3.3 - 3.4 would benefit from additional remarks or intuition around the theoretical results and the recursive analysis.

---

> ### Author Rebuttal · Authors · 2025-07-30
>
> We thank the reviewers for their thoughtful feedback. We are pleased that our work is recognized as technically rigorous and novel, and we appreciate the opportunity to clarify and address the concerns raised.
>
> 1. On the subsamplability parameter:
> The parameter m is unknown in general, as it encapsulates a property of the input distribution. However, if the data is known to come from a specific distribution (e.g., heavy tailed distributions), the value of m can often be computed analytically — see Section 4.1 and Appendix D for such examples. In the general case, while there is no universally optimal way to estimate m under privacy, one can still privately run the algorithm for multiple candidate values of m, since our method is differentially private for any choice of m. This pragmatic approach allows post-hoc validation of outputs and circumvents the need for stringent concentration bounds.Note that even "classical" results, such as estimating $\Sigma$ under the promise of data sampled from a Gaussian, make assumptions on the nature of the input and do not concern themselves with verifying whether such an assumption holds or not. The only difference is that we use a slightly less common assumption. But, as it turns out, our subsamplability assumption makes it possible to deal with both cases at the same time: (1) data drawn from a distribution and (2) data with a small fraction (roughly \beta/m) of outliers. Lastly, we ourselves tried to address the "minimality" of subsamplability in lines 25-29 in the introduction.
>
> 2. On the lack of experiments:
> Our work is entirely theoretical, and all claims are rigorously proven. Empirical experiments in such settings merely illustrate that the mathematical analysis holds — a fact already established through our theorems. In this light, we believe that including synthetic experiments would not significantly enhance the contribution, although we agree that future applied extensions could and should incorporate them.
>
>
> Specific comments for Reviewer #2:
>
> 1. On computational complexity:
> We agree that the algorithm might involve expensive spectral computations. However, similar limitations appear in prior work such as Kamath et al. (2019), which also refrain from approximating these steps. Replacing the decompositions with approximations is an interesting future direction, but outside the scope of our study, as it unclear how sensitive these approximation algorithms to the presence/omittion of one datapoint.
>
> 2. On sensitivity to R and \lambda_min:
> The bounds depend only logarithmically on these parameters (and thus polynomially in their natural representation). Thus, while important, they have limited effect on the overall accuracy guarantees.
>
> 3. On “slight violations” of the subsamplability condition:
> We would appreciate clarification on what is meant here. Obviously, if a requirement for our algorithm doesn't hold, then its guarantee is vacuous.
>
> 4. On the iteration bound and log base:
> The bound T=O(log(R/m)) is chosen for analytical tractability (as part of our preference to set parameters without using asymptotic notation). The base does not affect the asymptotic behavior; the important point is that the number of iterations depends logarithmically on the ratio R/m.
>
> 5. On estimating R:
> In practice, R can be chosen to include most of the data, using private range estimation techniques such as those in Nissim, Stemmer, and Vadhan (2016) or Nissim and Stemmer (2018).
>
> 6. On alternative sampling schemes:
> Sampling strategies like leverage-score sampling depend intricately on the data and are difficult to make private in a general-purpose way. Our result assumes uniform sampling. The term “any” in the title refers to input datasets satisfying the subsamplability assumption, not arbitrary sampling schemes - and as said earlier - these can be either inputs drawn from a distribution or some well-behaved input with a faction of outliers.
>
> 7. On manuscript structure:
> We thank the reviewer and welcome specific constructive suggestions to improve the clarity of Sections 3.3–3.4 and the introduction. We will incorporate more intuitive explanations and structural improvements in the revision. We just ask that the presentation suggestions will be agreed upon by all reviewers.

---

> > ### Comment · Reviewer_dJaK · 2025-08-06
> >
> > Nice theoretical work! Thank you for your response and clarification, which have addressed most of my concerns. I will give you an increase of 1 point.

---

### Official Review · Reviewer_AvYQ · 2025-07-08

**Clarity:** 3
**Significance:** 2
**Originality:** 3
**Rating:** 4
**Confidence:** 3

**Summary:**

This paper presents a novel differentially private algorithm for approximating the second-moment matrix of subsampleable datasets, even in the presence of outliers. By leveraging a recursive framework under zero-concentrated Differential Privacy, the authors achieve strong privacy-utility trade-offs and demonstrate robustness for heavy-tailed distributions and outlier-contaminated data. The theoretical contributions include formalizing subsamplability assumptions and providing concrete bounds for sample complexity and accuracy.

**Questions:**

1. In Theorem 3.1, what is the trade-off between privacy and estimation accuracy? Looks like the accuracy bound is only determined by $\alpha$.
2. Are there any empirical comparisons with existing methods to demonstrate the claimed advantages?

**Ethical Concerns:**

["NO or VERY MINOR ethics concerns only"]

**Final Justification:**

I will keep my positive score for potential acceptance.

**Limitations:**

See 'Weakness' part.

**Paper Formatting Concerns:**

No Formatting Concern

**Quality:**

3

**Strengths And Weaknesses:**

Strength:
1. The paper provides a comprehensive theoretical analysis, including formal proofs and clear assumptions, ensuring the robustness of the proposed algorithm.
2. The method achieves strong utility guarantees under a weaker subsamplability assumption compared to traditional concentration bounds, making it applicable to a broader range of datasets, including those with heavy-tailed distributions or outliers.

Weakness:
1. The paper lacks experimental results to validate the theoretical claims, making it difficult to assess practical performance.
2. The algorithm relies on multiple input parameters. How to tune these parameters and what is their impact on the estimation?

---

> ### Author Rebuttal · Authors · 2025-07-30
>
> We thank the reviewers for their thoughtful feedback. We are pleased that our work is recognized as technically rigorous and novel, and we appreciate the opportunity to clarify and address the concerns raised.
>
> 1. On the subsamplability parameter:
> The parameter m is unknown in general, as it encapsulates a property of the input distribution. However, if the data is known to come from a specific distribution (e.g., heavy tailed distributions), the value of m can often be computed analytically — see Section 4.1 and Appendix D for such examples. In the general case, while there is no universally optimal way to estimate m under privacy, one can still privately run the algorithm for multiple candidate values of m, since our method is differentially private for any choice of m. This pragmatic approach allows post-hoc validation of outputs and circumvents the need for stringent concentration bounds.Note that even "classical" results, such as estimating $\Sigma$ under the promise of data sampled from a Gaussian, make assumptions on the nature of the input and do not concern themselves with verifying whether such an assumption holds or not. The only difference is that we use a slightly less common assumption. But, as it turns out, our subsamplability assumption makes it possible to deal with both cases at the same time: (1) data drawn from a distribution and (2) data with a small fraction (roughly \beta/m) of outliers. Lastly, we ourselves tried to address the "minimality" of subsamplability in lines 25-29 in the introduction.
>
> 2. On the lack of experiments:
> Our work is entirely theoretical, and all claims are rigorously proven. Empirical experiments in such settings merely illustrate that the mathematical analysis holds — a fact already established through our theorems. In this light, we believe that including synthetic experiments would not significantly enhance the contribution, although we agree that future applied extensions could and should incorporate them.
>
>
> Specific comments for Reviewer #1:
> On baselines:
> Our comparison baseline provides only an \alpha-approximation of the second-moment matrix. This sets an initial threshold, whereas our algorithm achieves a \gamma-approximation for arbitrarily small \gamma, subject to sample complexity. Thus, the comparison is illustrative of a qualitative improvement in precision.

---

> > ### Comment · Reviewer_AvYQ · 2025-08-05
> > **Official Comment by Reviewer AvYQ**
> >
> > Thanks for the rebuttal, and I will keep my positive score for potential acceptance.

---

### Decision · Program_Chairs · 2025-09-17

**Decision:**

Accept (poster)

**Comment:**

The paper provides a differentially private algorithm for approximating a dataset's second moment matrix. When the data takes a certain ("subsamplable") form, they provide formal utility guarantees.

The reviewers generally appreciated the submission's algorithmic novelty and the basic conceptual idea of subsamplability, and all three recommended acceptance. However, all three reviewers also asked for clarifications regarding the "subsamplability" parameter and its effect on privacy (if I understand correctly, the assumption is only necessary for utility, not privacy), as well as the role of other input parameters; the authors should add discussion of those points to the final version.